# Myelin-reactive B cells exacerbate CD4+ T cell-driven CNS autoimmunity in an IL-23-dependent manner

Mohamed Reda Fazazi[1], Prenitha Mercy Ignatius Arokia Doss[1], Resel Pereira[2], Neva Fudge[3,4], Aryan Regmi[2,5], Charles Joly-Beauparlant[6], Irshad Akbar[1], Asmita Pradeep Yeola[1], Benoit Mailhot[1], Joanie Baillargeon[1], Philippe Grenier[6], Nicolas Bertrand[6,7], Steve Lacroix[1,8], Arnaud Droit [6,8], Craig S. Moore [3,4], Olga L. Rojas [2,5] & Manu Rangachari [1,8] ✉

B cells and T cells collaborate in multiple sclerosis (MS) pathogenesis. IgH[MOG] mice possess a B cell repertoire skewed to recognize myelin oligodendrocyte glycoprotein (MOG). Here, we show that upon immunization with the T cell-obligate autoantigen, MOG[35-55], IgH[MOG] mice develop rapid and exacerbated experimental autoimmune encephalomyelitis (EAE) relative to wildtype (WT) counterparts, characterized by aggregation of T and B cells in the IgH[MOG] meninges and by CD4+ T helper 17 (Th17) cells in the CNS. Production of the Th17 maintenance factor IL-23 is observed from IgH[MOG] CNS-infiltrating and meningeal B cells, and in vivo blockade of IL-23p19 attenuates disease severity in IgH[MOG] mice. In the CNS parenchyma and dura mater of IgH[MOG] mice, we observe an increased frequency of CD4+PD-1+CXCR5- T cells that share numerous characteristics with the recently described T peripheral helper (Tph) cell subset. Further, CNS-infiltrating B and Tph cells from IgH[MOG] mice show increased reactive oxygen species (ROS) production. Meningeal inflammation, Tph-like cell accumulation in the CNS and B/Tph cell production of ROS were all reduced upon p19 blockade. Altogether, MOG-specific B cells promote autoimmune inflammation of the CNS parenchyma and meninges in an IL-23-dependent manner.

Multiple sclerosis (MS) is a chronic autoimmune disease in which the adaptive immune system launches an attack against central nervous system (CNS) proteins, such as myelin. MS affects more than 2 million people worldwide[1]. Approximately 80% of patients present an initially relapsing-remitting (RR) disease course for which there are now more than 10 disease-modifying therapies available. However, 30–60% of these RR patients will eventually transition to a chronically worsening secondary progressive (SP) phase, characterized by gray matter

[1]axe Neurosciences, Centre de recherche du Centre hospitalier universitaire (CHU) de Québec – Université Laval, Pavillon CHUL, 2705 boul Laurier, Quebec City G1V 4G2 QC, Canada. [2]Krembil Research Institute, University Health Network, Toronto M5T 0S8 ON, Canada. [3]Division of BioMedical Sciences, Memorial University of Newfoundland, St. John's, NL A1B 3V6, Canada. [4]Department of Neurology, Faculty of Medicine, Memorial University of Newfoundland, St. John's, NL A1B 3V6, Canada. [5]Department of Immunology, University of Toronto, Toronto M5S 1A1 ON, Canada. [6]axe Endocrinologie et nephrologie, Centre de recherche du Centre hospitalier universitaire (CHU) de Québec – Université Laval, Pavillon CHUL, 2705 boul Laurier, Quebec City, QC G1V 4G2, Canada. [7]Faculty of Pharmacy, Laval University, 1050 ave de la Médecine, Quebec City, QC G1V 4G2, Canada. [8]Department of Molecular Medicine, Faculty of Medicine, Laval University, 1050 ave de la Médecine, Quebec City, QC G1V 4G2, Canada. ✉e-mail: manu.rangachari@crchudequebec.ulaval.ca

pathology and for which treatment options are limited[2]. Pathophysiological mechanisms in progressive MS are thus of intense current interest[3].

CD4[+] T cells have historically been the most intensively studied players in the immune pathogenesis of MS. However, it has become increasingly clear that B cells additionally play important roles in MS. Clonally expanded B cells are present in the cerebrospinal fluid (CSF) and MS plaques[4–6], and the presence of meningeal lymphocytic follicles adjacent to cortical lesions is associated with disease progression[7,8]. Further, antibodies against myelin oligodendrocyte glycoprotein (MOG), a key component of myelin, were found in active MS lesions[9]. Crucially, the B cell-depleting anti-CD20 therapies rituximab[10], ocrelizumab[11] and ofatumumab[12] improve RRMS, and ocrelizumab is the only FDA-approved drug for primary progressive MS[13]. Curiously, though, CD20 is not expressed by antibody-secreting plasmablasts and plasma cells[14]. This suggests that the pathogenic role of B cells in MS might lie less with the generation of autoantibodies, and more with their capacity to produce cytokines and to interact with other immune cell types, such as T cells[15].

Experimental autoimmune encephalomyelitis (EAE) is an animal disease that recapitulates many of the immune aspects of MS pathogenesis. Use of this model has helped us to understand the role of T cells, and CD4[+] T cells in particular, in the initiation and maintenance of autoreactive inflammation in the CNS[16]. However, studies using the transgenic IgH[MOG] mouse strain have indicated that B cells may also play a crucial role in EAE pathology[17–19]. These mice (also known as Th) express a knocked-in IgH chain derived from a MOG-specific antibody; thus, around 30% of their B cells are therefore specific for MOG protein[17]. IgH[MOG] animals develop severe EAE when immunized with either whole MOG protein[17] or with its extracellular domain (MOG[1-125])[20], indicating an important role for MOG-reactive B cells in neuroimmune processes. Further, they develop spontaneous EAE when crossed to myelin peptide-specific T cell receptor transgenic lines on the C57BL/6J[18] and SJL/J[19] backgrounds. Indeed, while T cell help is required for the full activation of B cells in the majority of cases, there is increasing evidence that B cells can reciprocally promote effector T cell responses, notably those of the Th17 lineage[21–25]. However, the potential mechanisms by which MOG-reactive B cells facilitate T cell-driven pathogenicity, such as in class II-restricted peptide immunization models of EAE, remain incompletely understood.

Here, we studied the co-operative role of B cells and T cells in CNS autoimmunity using IgH[MOG] mice on the non-obese diabetic (NOD) genetic background on which EAE develops with a relapsing/chronic disease pattern[26]. When immunized with the T cell-obligate peptide MOG[35-55], IgH[MOG] mice developed a rapid, severe form of EAE characterized by meningeal clusters of CD4[+] T cells, B cells and CD11c[+] dendritic cells (DCs) that displayed an intricate network in the underlying extracellular matrix (ECM) that was reminiscent of organized tertiary lymphoid organs (TLOs). Th17 responses were specifically upregulated in the inflamed IgH[MOG] CNS and, surprisingly, CNS-infiltrating B cells produced the Th17 maintenance factor IL-23. Further, PD1[+]CXCR5[-]CD4[+] T cells were present at a high frequency in the IgH[MOG] CNS parenchyma and dura mater; these cells resembled T peripheral helper (Tph) cells, a recently described effector T cell subset that associates with B cells in the inflamed synovium in rheumatoid arthritis (RA)[27]. Transcriptomics analysis of CNS-infiltrating B and Tph-like cells revealed that genes related to pathways of neurodegeneration and oxidative phosphorylation were upregulated in both cell types in IgH[MOG] mice, suggesting that these processes contribute to the heightened pathology observed in these animals. Notably, IL-23 blockade abrogated severe disease in IgH[MOG] mice, as well as both CNS Tph-like cell accumulation and ROS production from both B cells and Tph-like cells. Together, our results demonstrate that MOG-specific B cells play a crucial role in augmenting CD4[+] T cell-driven EAE in an IL-23-dependent manner.

## Results

### IgH[MOG] mice develop severe EAE upon active immunization with MOG[35-55]

When immunized with MOG[35-55], wildtype (WT) NOD-background mice display a disease course characterized by relapse-remitting disease in the early phase that transitions to a chronic worsening phase in some animals[26]. NOD-EAE has thus been considered a possible model of SPMS[28,29], though others argue against this interpretation[30]. We confirmed that MOG[35-55] immunized WT NOD mice develop disease characterized by relatively mild symptoms over the first ~100 days (Supplementary Fig. 1a). We next compared the development of EAE between NOD-background WT and IgH[MOG] mice upon MOG[35-55] immunization. Both male and female IgH[MOG] mice developed severe disease within 25 days (Fig. 1a), with a substantial frequency of these mice attaining ethical endpoints (6/8 IgH[MOG] males vs. 0/8 WT males, p = 0.007; 7/10 IgH[MOG] females vs. 0/10 WT females, p = 0.0031). Thus, throughout the remainder of the study, we used both male and female animals. Histopathological analyses revealed increased lymphocyte infiltration and demyelination in the spinal cords of IgH[MOG] mice relative to controls (Fig. 1b, Supplementary Fig. 1b).

We next examined peripheral T cell responses prior to disease onset. No differences in IL-17 or IFNγ were detected by either ex vivo flow cytometry (Fig. 1c) or by ELISA after MOG[35-55] peptide recall (Supplementary Fig. 1c). Next, to identify a possible role for IgH[MOG] B cells in directly driving T cell responses, we pulsed IgH[MOG] or WT B cells with MOG[35-55] and co-cocultured them with MOG[35-55]-specific 1C6 transgenic T cells[31–33]. However, no differences were observed in T cell proliferation (Fig. 1d), indicating that IgH[MOG] B cells were not intrinsically better at presenting peptide antigen to T cells. Furthermore, while the cervical LNs (cLN) are a potential site of T cell reactivation in MS[34], we observed only limited production of IL-17 and IFNγ in this compartment; while a large proportion of CD4[+] T cells in cervical LNs were positive for TNF, no differences were detected between WT and IgH[MOG] (Supplementary Fig. 1d).

Antigen-specific antibody (Ab) secretion is the primary function of B cells. Oligoclonal immunoglobulin (Ig) banding in cerebrospinal fluid (CSF) is an important diagnostic marker for MS[35] and it has been suggested that the production of IgG anti-MOG autoAbs by IgH[MOG] B cells facilitates rapid and severe EAE[36]. However, no differences in MOG-specific circulating IgG were found in IgH[MOG] serum relative to controls (Fig. 1e), suggesting that severe disease in transgenic animals was not accompanied by an increase in MOG-specific autoantibodies in the peripheral circulation. Together, these data showed that the presence of myelin-reactive B cells can exacerbate CNS autoimmunity and tissue damage in CD4[+] T cell-driven EAE. However, increased disease severity did not appear to be accompanied by augmented production of inflammatory cytokines by peripheral T cells, nor by an increase in circulating MOG-specific autoAbs.

### Accumulation of meningeal TLOs in IgH[MOG] mice

Target organ-infiltrating immune cells are essential to the pathogenesis of autoimmune disease. We therefore enumerated the frequency and absolute number of CNS-infiltrating immune cells in immunized WT and IgH[MOG] mice. The relative frequencies and absolute numbers of CD4[+] T cells, CD8[+] T cells, CD11c[+] dendritic cells (DCs), CD11b[+]CD11c[-] macrophages and Ly6G[+] neutrophils were significantly increased in the CNS of IgH[MOG] mice relative to WT; however, B cell frequency and absolute numbers were unchanged (Supplementary Fig. 2a).

Meningeal TLOs have been documented in progressive MS and their presence correlates with poor outcomes[7,37]. The presence of analogous structures in EAE is dependent on murine genetic background and the manner of disease induction; while TLO are observed upon active immunization of SJL/J mice with PLP[139-151] peptide[38], they do not appear in MOG[35-55]-immunized C57BL/6J mice, rather arising only upon adoptive transfer of MOG[35-55]-specific Th17 cells[39]. Whether or not they

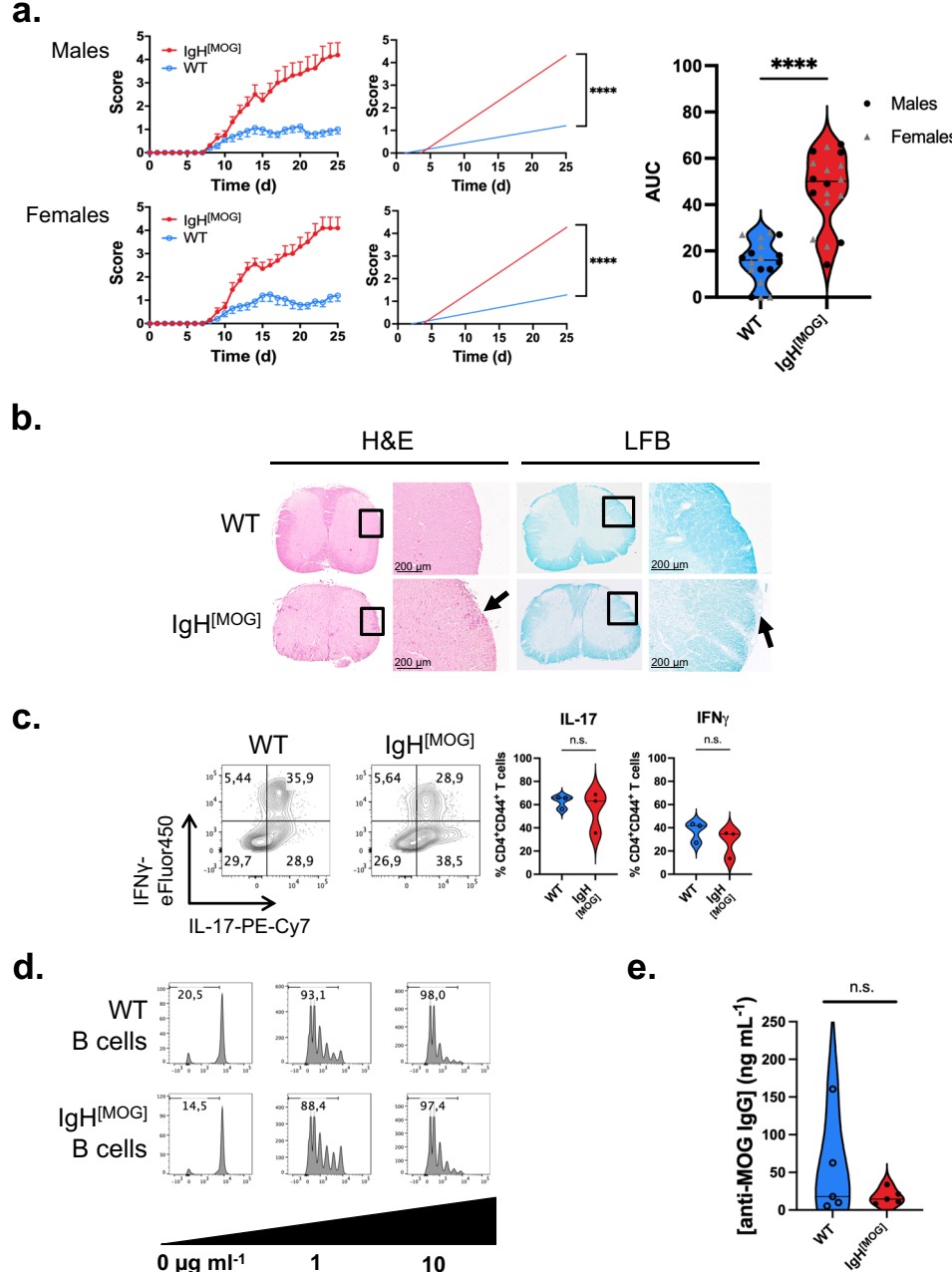

**Fig. 1 | Active immunization with MOG$_{[35-55]}$ induces severe EAE in IgH$^{[MOG]}$ mice.**
**a** *Left*, Disease curves and linear regression of males (WT, $n = 8$, IgH$^{[MOG]}$, $n = 8$) and females (WT, $n = 10$, IgH$^{[MOG]}$, $n = 10$) immunized with MOG$_{[35-55]}$ and monitored for development of EAE. Linear regression was used for statistical analysis. Representative of > 20 experiments conducted in each sex. *Right*, area under curve (AUC) analysis of disease curves in left panels. Two-way ANOVA was used to assess the contribution of genotype. The contribution of sex was $p < 0.9912$ and there was no interaction effect between the variables. **b** Spinal cords from female WT and IgH$^{[MOG]}$ mice at endpoints were sectioned and stained with hematoxylin & eosin (H&E; immune infiltration) and Luxol fast blue (LFB; myelin). 10X magnification. Representative of $n = 4$ mice of each genotype from a single immunization. Quantification in Supplementary Fig. 1b. **c** Draining LN cells were isolated at disease onset from female WT ($n = 3$) and IgH$^{[MOG]}$ ($n = 3$) mice from a single immunization

and IFNγ and IL-17 expression were determined by flow cytometry. Gated on live CD4$^+$CD44$^+$ events. Two-tailed *t*-test was used. **d** Female WT or IgH$^{[MOG]}$ B cells were co-cultured with CellTrace Violet-labeled MOG$_{[35-55]}$-specific 1C6 T cells that were pulsed, or not, with the indicated concentrations of MOG$_{[35-55]}$. Gate frequencies indicate the percentage of cells undergoing at least one cell division. Representative of cultures derived from individual mice ($n = 3$ mice of each genotype). Data is from one experiment. **e** Sera were collected from male WT ($n = 5$) and IgH$^{[MOG]}$ ($n = 5$) mice at endpoints and the concentration of MOG-specific IgG was assessed by ELISA. Representative of one of two immunizations. A two-tailed *t*-test was used. In all violin plots, horizontal lines represent medians. Each datapoint represents an individual mouse. Exact *p*-values can be found in the Source Data file. Linked to Supplementary Fig. 1.

are a feature of NOD-EAE is currently unknown. As B cells are crucial to TLO formation[39], we assessed their presence in the decalcified brains of immunized WT versus IgH$^{[MOG]}$ mice. Strikingly, we observed clustering of CD4$^+$ T cells, B cells and CD11c$^+$ DCs of varying magnitude in the CNS of IgH$^{[MOG]}$ mice, but only rarely in WT mice (Fig. 2a). These were

observed in the meninges of the brain, the brain sulci and the cerebellar meninges (Supplementary Fig. 2b). Furthermore, in the brain meninges, we found an organized expansion of fibronectin and PDGFRα/β underneath the B cell clustering (Fig. 2b); this was indicative of the stroma cell network that characterizes TLOs[39,40].

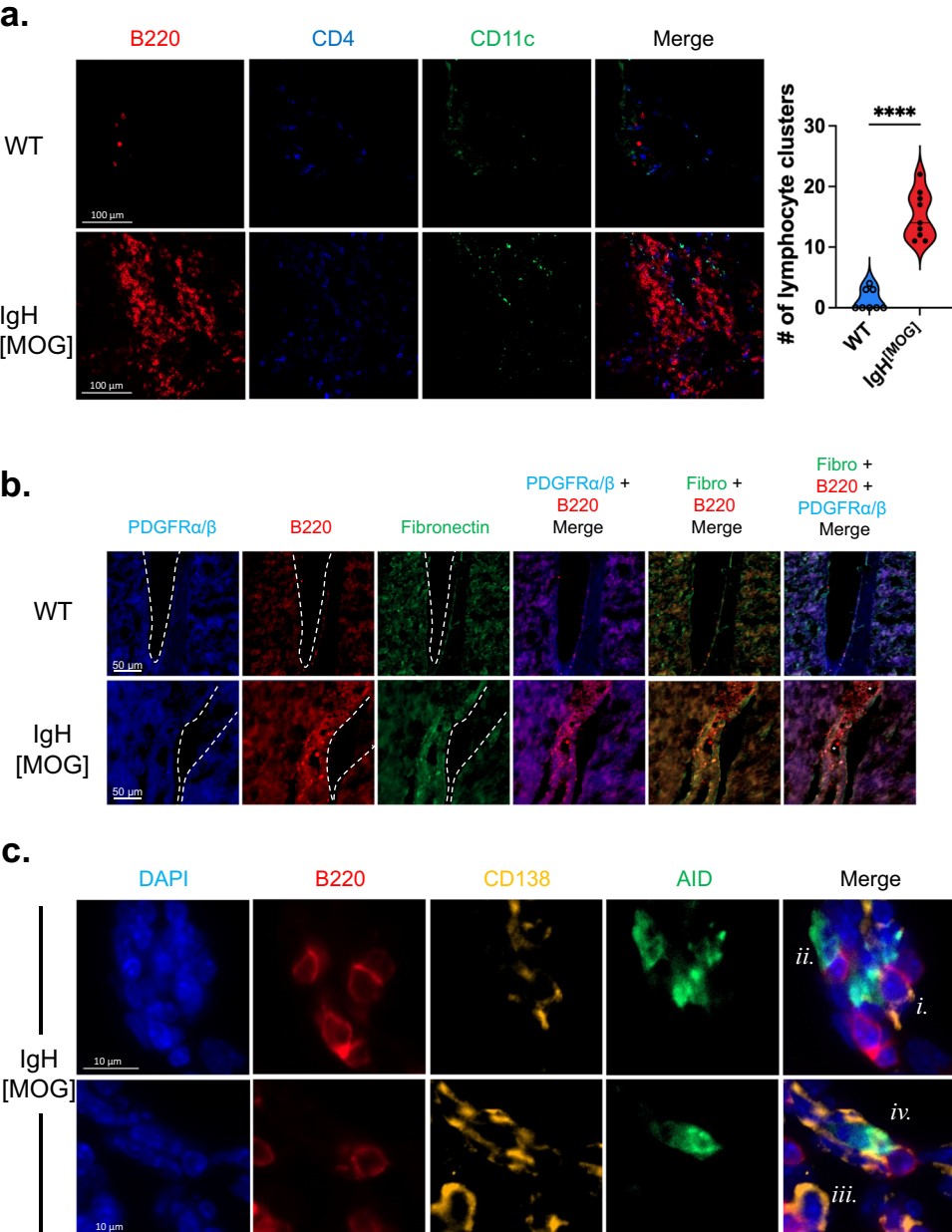

**Fig. 2 | Increased lymphocytic aggregation in the meninges of immunized IgH[MOG] mice. a** Brain and cerebellar meningeal sections from female MOG[35-55]-immunized WT (n = 8) or IgH[MOG] (n = 9) mice were assessed by immunofluorescence (IF) for expression of B220 (red), CD4 (blue), and CD11c (green) at endpoints, and lymphocyte aggregates were enumerated. Data are from two immunizations. Two-tailed t-test was used. In violin plot, horizontal line represents median. Each datapoint represents an individual mouse. **b** Assessment of the ECM near meninges in the cerebellum was determined by IF. Expansion on the meningeal ECM is shown by fibronectin (green) and PDGFRα/β (blue) which overlap with B220 (red) B cell clusters. Dotted lines outline the meningeal membrane and (*) denotes vascular endothelium. Representative of n = 4 female mice of each genotype. Data are from two immunizations. **c** Representative IF images of IgH[MOG] meningeal lymphocytic clusters, stained for CD138 (yellow), AID (green), B220 (red), DAPI (blue). *i.*, CD138+B220+, *ii.*, CD138-AID+, *iii.*, CD138+B220-, *iv.*, CD138+AID+. The source meningeal cluster is presented in Supplementary Fig. 2c. Representative of 4 female mice. Data are from two immunizations. Exact p-values can be found in the Source Data file. Linked to Supplementary Fig. 2.

We next assessed markers of B cell activation in the brain and associated tissue of immunized IgH[MOG] mice. Within areas of B cell accumulation in the meninges, we observed CD138+ plasma cells (Fig. 2c, Supplementary Fig. 2c); as described previously, these cells were either B220-positive or B220-negative[41,42]. We additionally found cells positive for activation-induced cytidine deaminase (Fig. 2c, Supplementary Fig. 2c), an enzyme required for somatic hypermutation and class-switching[43]. Finally, we observed IgG deposition in the brain parenchyma of immunized IgH[MOG] mice, with significantly higher levels compared to WT animals (Supplementary Fig. 2d). These data

therefore point to the accumulation of lymphocyte clusters in the meninges of MOG[35-55]-immunized IgH[MOG] mice, with features reminiscent of those seen in TLOs.

## The IgH[MOG] CNS is characterized by inflammatory B cells and Th17 cells

We next investigated whether there might be differences in the proportion of functional B cell subsets between immunized WT and IgH[MOG] mice. Indeed, we observed a stark increase in the frequency of class-switched (CS) IgM-IgD- B cells[44] in the CNS (Fig. 3a) of IgH[MOG]

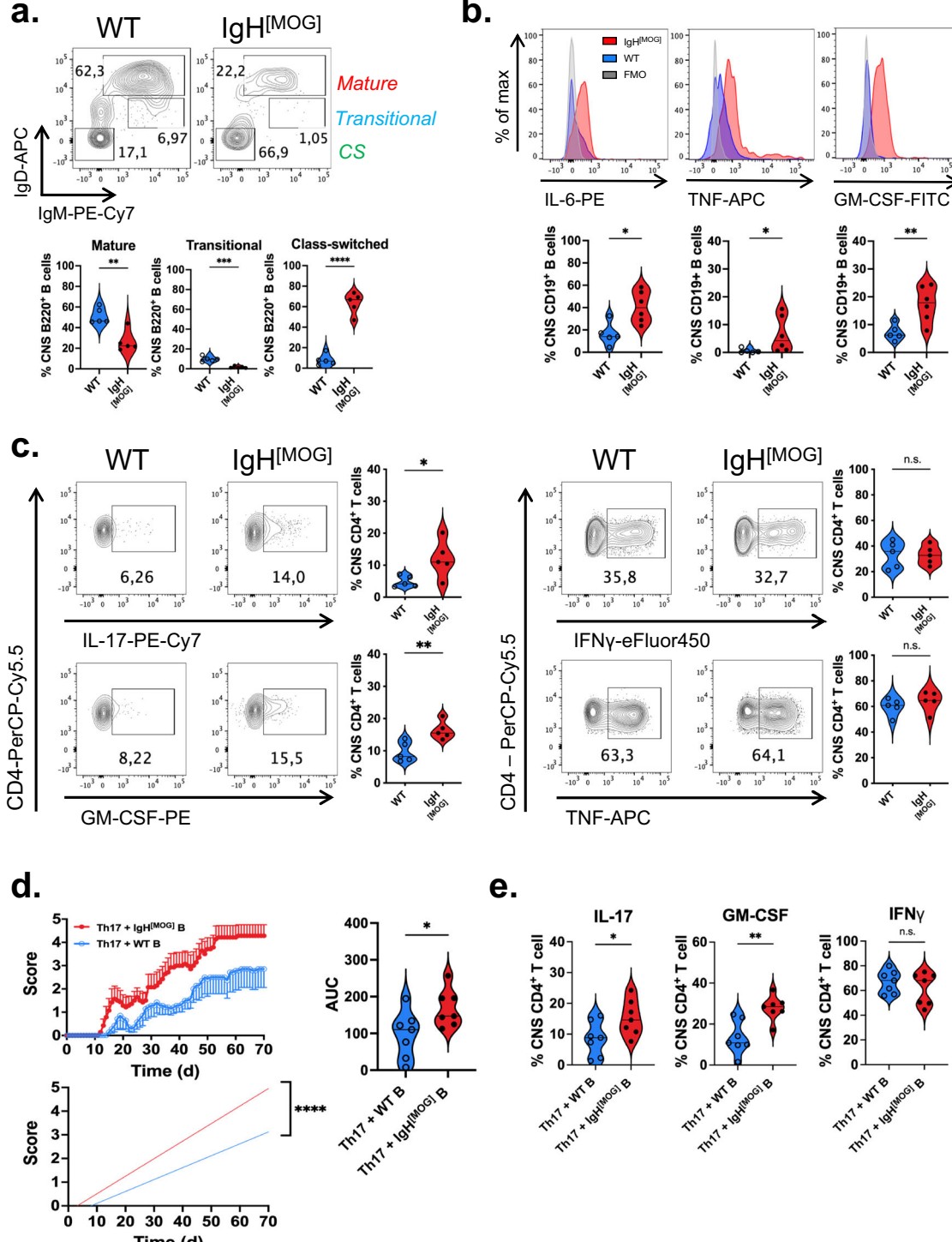

**Fig. 3 | Inflammatory B cells facilitate Th17 responses in IgH[MOG] mice. a** The frequency of mature (IgM[hi]IgD[hi]), transitional (IgM[hi]IgD[mid]) and class-switched (CS; IgM[-]IgD[-]) B cells were assessed from female WT (n = 5) and IgH[MOG] (n = 5) CNS at endpoints. Gated on live B220[+] events. Representative of one of two immunizations. Two-tailed t-test was used. **b** Expression of the indicated cytokines was assessed from female WT (n = 5) and IgH[MOG] (n = 6) CNS B cells at endpoints. Gated on live CD19[+] events. Representative of one of two immunizations. Two-tailed t-test was used. **c** Female WT (n = 5) and IgH[MOG] (n = 5) were immunized with MOG[35-55] and CNS-infiltrating CD4[+] T cells were isolated at endpoints. Expression of the indicated cytokines was assessed by intracellular flow cytometry. Gated on live CD4[+] events. Representative of one of two immunizations. Two-tailed t-test was

used. **d** Female NOD.*Scid* mice were passively infused with 2 × 10[6] female WT or IgH[MOG] B cells, and received 2×10[6] 1C6 Th17 cells 7 days later. Mice were monitored for signs of EAE. *Left*, disease curve and linear regression from one of two transfer experiments; *right*, AUC comparison of mice from both transfers (n = 7, both groups of recipient mice). Two-tailed t-test was used. **e** CNS-infiltrating CD4[+] T cells were isolated from adoptive transfer recipients in (**d**) at endpoints and the indicated cytokines were assessed by flow cytometry. Two-tailed t-test was used. Representative plots for data in Fig. 3e are presented in Supplementary Fig. 3g. In all violin plots, horizontal lines represent medians. Each datapoint represents an individual mouse. Exact p-values can be found in the Source Data file. Linked to Supplementary Fig. 3.

mice, which was accompanied by a decrease in the frequency of transitional and mature B cells[45–47] relative to WT counterparts. In line with our finding that serum MOG-specific IgG was comparable between WT and IgH[MOG], we observed no differences in B cell subpopulation frequency in the spleens of immunized mice, with the frequency of CS B cells (<5%) being negligible in both genotypes (Supplementary Fig. 3a). Interestingly, a comparison of splenic B cell subsets in unimmunized mice revealed differences in the proportion of mature and transitional B cells; the former were decreased, and the latter increased, in IgH[MOG] mice. Again, however, the frequency of CS B cells was <5% in both cases (Supplementary Fig. 3b).

B cells generate canonical inflammatory cytokines such as IL-6, TNF and GM-CSF, with their production of these cytokines being linked to worsened outcomes in MS[15,48,49]. We observed that a significantly greater proportion of IgH[MOG] CNS B cells produced these cytokines as compared to WT counterparts (Fig. 3b). While we observed no difference in MHC class II expression between CNS-infiltrating WT and IgH[MOG] B cells (Supplementary Fig 3c), an increase in GL-7+Fas+ germinal center (GC) B cells was seen in the IgH[MOG] CNS (Supplemental Fig. 3d).

The IgH[MOG] B cell repertoire features a substantial proportion of MOG-nonspecific clones[17] and it was previously reported that MOG-reactive B cells are excluded from the CNS[50,51]. We therefore interrogated the specificity to MOG antigen of B cells from WT and IgH[MOG] mice in peripheral immune and CNS tissue. As expected[17], a robust proportion of IgH[MOG] splenic B cells were MOG-specific; this was also the case in IgH[MOG] cLN that drain the CNS. By contrast, WT B cells from these compartments were MOG-non-specific (Supplementary Fig 3e). With respect to the CNS, and in line with previous observations[50,51], the IgH[MOG] CNS parenchyma and subdural meninges (SDM) were devoid of MOG-specific B cells. Intriguingly, however, we observed the presence of MOG-specific B cells in the dura mater of immunized IgH[MOG] mice (Supplementary Fig 3f). Thus, the B cell repertoire in the CNS-associated tissue of IgH[MOG] mice is predominantly, but not exclusively, MOG-non-specific.

As we had observed an elevated frequency of CD4+ T cells in the CNS of sick IgH[MOG] mice, we examined the capacity of these cells to produce inflammatory Th1 and Th17 cytokines by flow cytometry, due to the well-established role of these CD4+ effector T cell subsets in EAE[52]. We observed no differences between IgH[MOG] and WT CD4+ T cells in their production of IFNγ and TNF in the CNS. By contrast we saw an upregulation of IL-17 production from CNS-infiltrating CD4+ T cells from IgH[MOG] at disease endpoint (Fig. 3c). Notably, production of GM-CSF, a key pathogenic cytokine implicated in Th17-driven tissue inflammation, was also augmented in IgH[MOG] CD4+ T cells.

B cells have been reported as promoting Th17 differentiation[22]; further, both the congenital absence[23] and therapeutic depletion[24,25] of B cells correlates with diminished circulating Th17 cell frequency. Thus, to test whether IgH[MOG] B cells facilitate Th17 responses in vivo, we adapted our previously described Th17 adoptive transfer protocol[33] by seeding NOD.Scid mice with immunologically naïve WT or IgH[MOG] B cells, and subsequently injecting MOG[35-55]-specific 1C6 transgenic[31] Th17 cells. Mice that had been infused with IgH[MOG] B cells developed substantially more severe Th17-driven EAE than those reconstituted with WT B cells (Fig. 3d). This was characterized by an increased frequency of IL-17+ and GM-CSF+ CNS-infiltrating Th17 cells in IgH[MOG] B cell-reconstituted hosts (Fig. 3e, Supplementary Fig. 3g). Taken together, these data show that severe EAE in IgH[MOG] mice is characterized by the presences of inflammatory B cells and Th17 cells, and that IgH[MOG] B cells can promote Th17-mediated disease processes.

## Augmented IL-23 production by IgH[MOG] mice leads to exacerbated disease

These findings led us to investigate whether IgH[MOG] B cells may be characterized by increased production of the Th17 stabilization factor IL-23, which is essential to Th17 pathogenicity in vivo[53,54] and which can be secreted by human B cells[55]. Indeed, we found that CNS IgH[MOG] B cells show higher expression of IL-23 heterodimer relative to WT counterparts (Fig. 4a). In the spleen, IL-23 expression from both WT and IgH[MOG] B cells was modest and no differences were noted between the genotypes (Supplementary Fig. 4a); curiously, IL-23 expression was higher in MOG-nonspecific splenic B cells from IgH[MOG] mice as opposed to the MOG-specific proportion (Supplementary Fig. 4b). Both WT and IgH[MOG] B cells were negative for the p35 subunit of the Th1 differentiation factor IL-12 (Supplementary Fig. 4c), which shares the p40 subunit with IL-23. No differences in IL-23 heterodimer positivity were detected between CNS-infiltrating WT and IgH[MOG] DCs, macrophages and microglia (Supplementary Fig. 4d). Notably, immunofluorescence (IF) staining showed co-localization of IL-23 with B cells in or near lymphocyte aggregates in the IgH[MOG] meninges (Fig. 4b).

The robust upregulation of IL-23 by B cells in the IgH[MOG] CNS led us to ask whether this cytokine might be essential for the exacerbated disease seen in these animals. We therefore actively immunized IgH[MOG] mice and administered either anti-IL-23p19 blocking antibody[56] or isotype control. In vivo blockade abrogated severe EAE (Fig. 4c), characterized by reduced lymphocyte infiltration and demyelination in the spinal cord (Supplementary Fig. 4e). As Th17 responses are crucial to TLO formation[39], we next examined the effects of IL-23 blockade on the presence of meningeal lymphocytic aggregation. Administration of anti-IL23p19 diminished the number of lymphocyte clusters in the meninges of IgH[MOG] mice (Fig. 4d); concomitantly, we observed a decrease in the underlying ECM network (Supplementary Fig. 4f).

Turning to the impact of IL-23 in cellular inflammation in the IgH[MOG] CNS, we found that the frequency of class-switched B cells was reduced in immunized animals treated with anti-p19 (Fig. 4e). Further, CNS-infiltrating CD4+ T cells from anti-p19-treated IgH[MOG] mice showed decreased IL-17 expression and a trend towards reduced GM-CSF (Fig. 4f); however, no differences in IFNγ production between T cells from anti-19 or isotype-treated mice were observed.

The cerebrospinal fluid is a useful proxy for pathologic processes within the CNS parenchyma[57] and is routinely sampled in MS diagnosis. We therefore asked whether expression of IL-23p19 transcript in the CSF of people with MS (pwMS) were altered in relation to the presence of B cells (Supplementary Table 1). We examined the CSF of 11 pwMS for expression of IL23A (IL-23p19 gene name) and correlated this to the frequency of CD19+ B cells in the CSF of the same individuals. A robust correlation was identified (Spearman $r = 0.7107$) between IL23A and the proportion of CD45+ leukocytes that were CD19+ (Fig. 4g). We then analyzed publicly available bulk RNA-seq data derived from peripheral B cells of pwMS versus healthy controls[58], and observed a trend towards increased IL23A expression in the former (Supplementary Fig. 4g). In sum, these data show that excessive production of IL-23 by IgH[MOG] B cells may underpin the severe pathology observed in these animals and that expression of IL-23 transcript is associated with B cells from pwMS.

## Tph-like cells accumulate in the IgH[MOG] CNS

T follicular helper (Tfh) cells play a critical role in GC formation and B cell immune activation, and were recently shown to collaborate with B cells in driving Th17-dependent EAE[44]. We therefore examined the presence of PD-1+CXCR5+ Tfh cells in splenic and CNS tissues from WT and IgH[MOG] mice. Intriguingly, while Tfh were relatively rare in all cases, we instead observed an increase in PD-1+CXCR5- CD4+ T cells in the IgH[MOG] CNS (Fig. 5a). These cells were reminiscent of the recently discovered Tph cell subset[27,59]. In contrast to the CNS, we observed no differences in the frequency of PD-1+CXCR5- CD4+ T cells in the spleen (Supplementary Fig. 5a). Further, we found evidence of CXCR5 positivity in the CD4neg CNS cell fraction (Supplementary Fig. 5b), indicating that Collagenase D digestion did not abrogate surface expression of this marker[60].

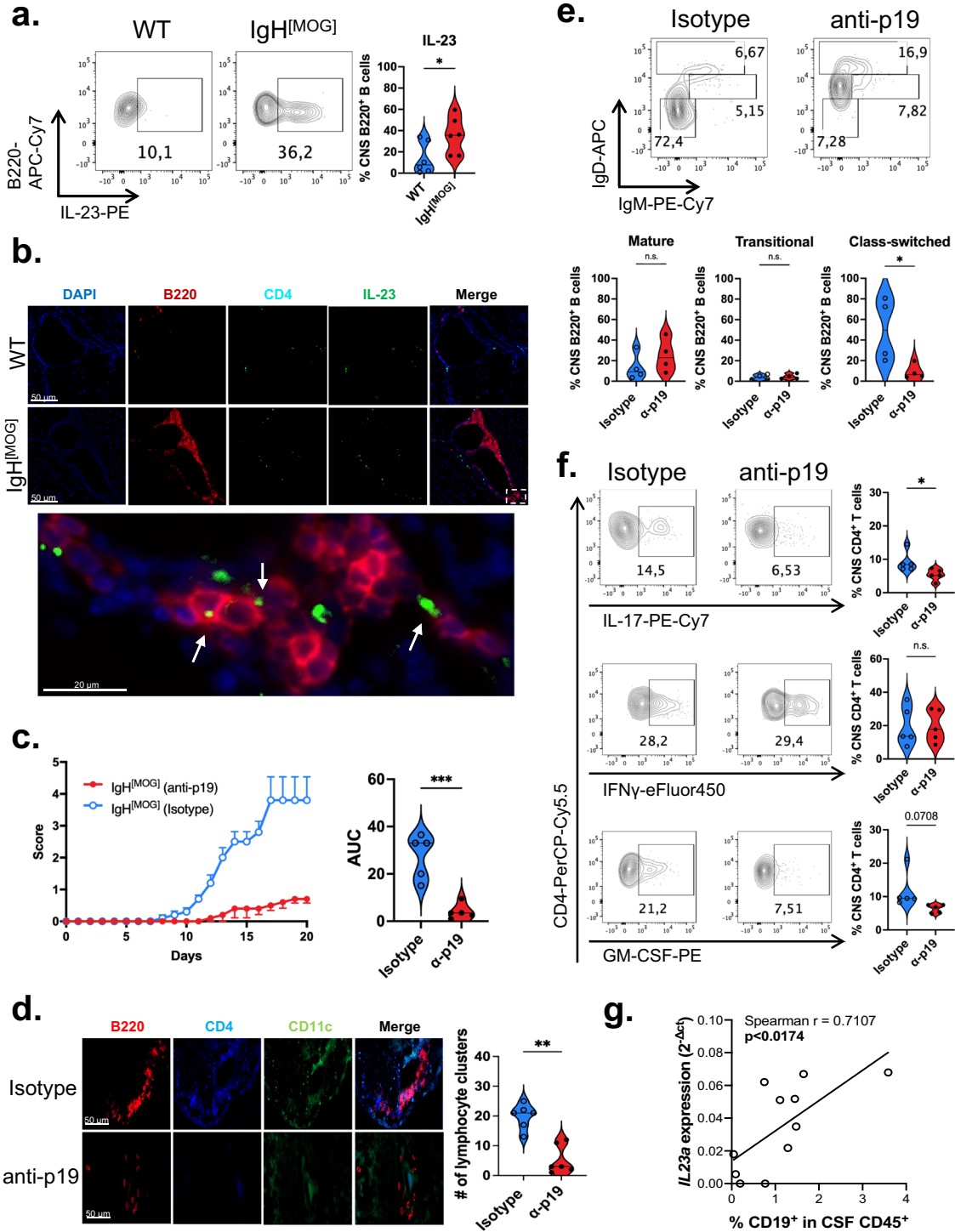

**Fig. 4 | CNS IgH[MOG] B cells upregulate IL-23. a** Male WT ($n = 6$) and IgH[MOG] ($n = 6$) mice were immunized with MOG[35-55] and B cells were isolated from CNS tissues at endpoints. Expression of IL-23 heterodimer was assessed by intracellular flow cytometry. Data are presented from one of four immunizations. Two-tailed $t$-test was used. **b** IF detection of IL-23 (green), B220 (red), CD4 (aqua), DAPI (blue) in the meninges of female MOG[35-55]-immunized WT or IgH[MOG] mice. Representative of $n = 3$ mice of each genotype from a single immunization. **c** Female IgH[MOG] mice were immunized with MOG[35-55] and were treated on day (-1) and day 6 with 1 mg anti-p19 ($n = 5$) or rIgG1 isotype ($n = 5$). Disease curves representative of one of seven experiments. Two-tailed $t$-test was used. **d** Cerebellar meningeal sections from MOG[35-55]-immunized female IgH[MOG] mice, treated with isotype ($n = 6$) or anti-p19 ($n = 6$), were assessed for lymphocyte aggregation. Data are from two immunizations. Two-tailed $t$-test was used. **e** CNS-infiltrating mature, transitional

and class-switched B cells were assessed by flow cytometry from anti-p19 ($n = 4$) or rIgG1 isotype ($n = 4$)-treated immunized female IgH[MOG] mice. Gated on live B220+ events. Data are from a single immunization. Two-tailed $t$-test was used. **f** CD4+ T cells were isolated from p19 ($n = 5$) or rIgG1 isotype ($n = 5$)-treated immunized female IgH[MOG] mice and the indicated cytokines were assessed by flow cytometry. Data are representative of one of two immunizations. Two-tailed $t$-test was used. **g** IL23A was assessed from CSF immune cells of MS-affected individuals (see individual information on pwMS in Supplementary Table 1) by qPCR and was correlated to the frequency of CD19+ B cells within the CD45+ CSF leukocyte population of each patient. Linear regression analysis was applied to calculate Spearman r and $p$-value. In all violin plots, horizontal lines represent medians. Each datapoint represents a biological replicate. Exact $p$-values can be found in the Source Data file. Linked to Supplementary Fig. 4.

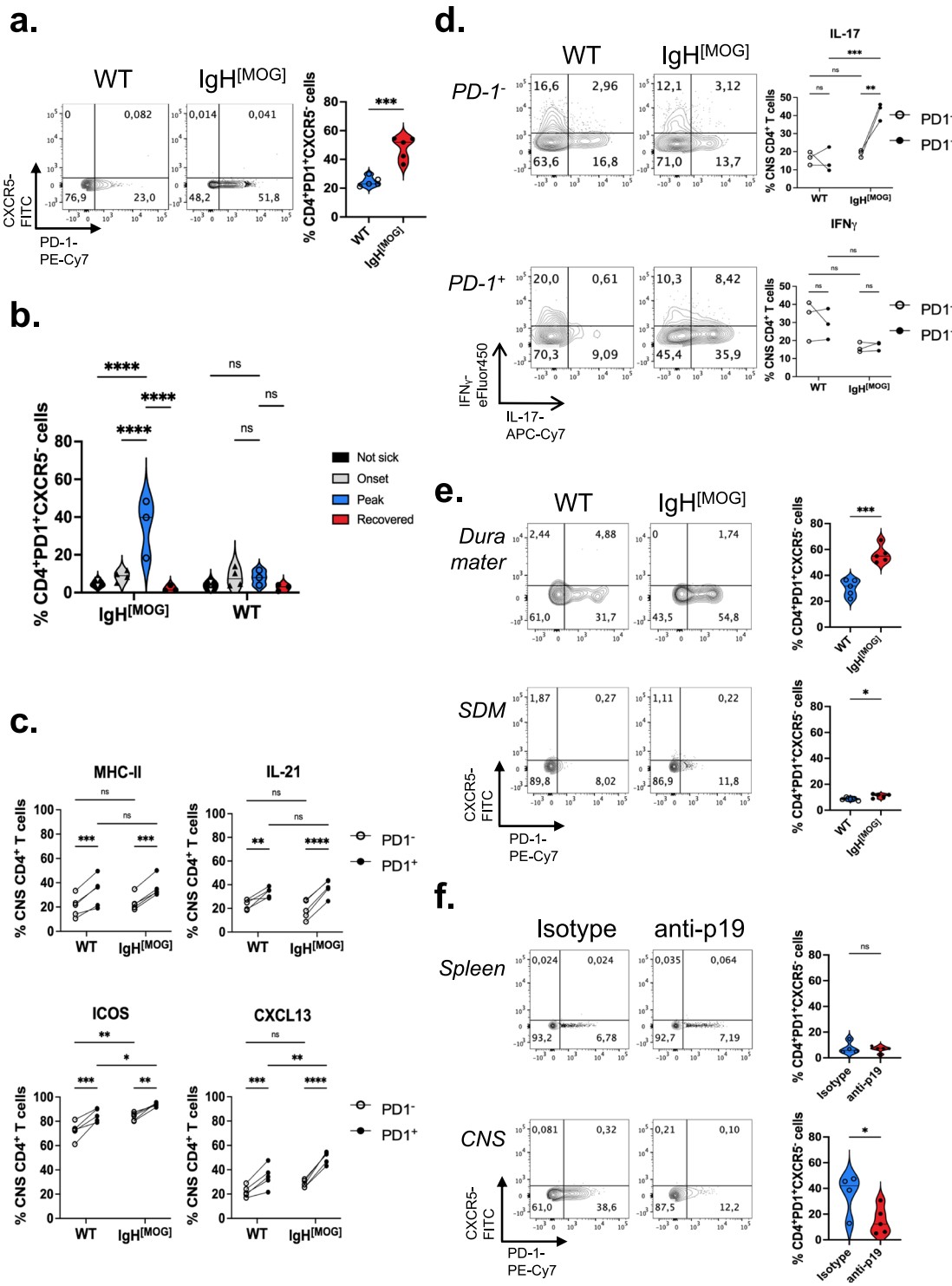

Originally discovered in the context of rheumatoid arthritis (RA), Tph cells are recruited to the inflamed synovium, where they associate with B cells. They are distinct from Tfh[27], and indeed we found that PD-1[+]CXCR5[-] cells from both WT and IgH[MOG] mice were negative for Bcl6 (Supplementary Fig. 5b), the Tfh master transcription factor[61]. While present in human RA[27,59] and lupus[62], evidence for the accumulation of Tph cells in murine autoimmune disease is, thus far, limited. Interestingly, the frequency of PD-1[+]CXCR5[-] CD4[+] T cells was higher at the peak of EAE in IgH[MOG] mice as opposed to at onset or recovery (Fig. 5b).

MHC class II, IL-21, ICOS and CXCL13 are all highly expressed on Tph cells[27]. We therefore compared expression of these markers between CNS-infiltrating PD-1[neg] and PD-1[+] CD4[+] T cells of both WT and IgH[MOG] mice. In both genotypes, expression of all four markers was upregulated in PD-1[+] cells, relative to PD-1[neg] cells (Fig. 5c, Supplementary Fig. 5c). Curiously, CXCL13 was specifically upregulated by IgH[MOG] PD-1[+] T cells relative to WT PD-1[+] T cells, yet not by IgH[MOG] PD-1[neg] cells relative to WT PD-1[neg] T cells (Fig. 5c, Supplementary Fig. 5c). IF analysis of IgH[MOG] meningeal tissue further revealed the presence of CXCL13-positive CD4[+] T cells (Supplementary Fig 5d); thus adding to the accumulating evidence that murine CD4[+] T cells can express CXCL13[63,64]. In spleens, expression of MHC class II, ICOS and CXCL13 were again increased in PD-1[+] versus PD-1[neg] CD4[+] T cells of both

**Fig. 5 | Tph cells infiltrate the IgH[MOG] CNS. a** Male WT (*n* = 5) and IgH[MOG] (*n* = 5) mice were immunized with MOG[35-55] and T cells were isolated from CNS at endpoints. Expression of PD-1 and CXCR5 was assessed by flow cytometry. Data are representative of seven experiments. Two-tailed *t*-test was used. **b** Male WT or IgH[MOG] mice were immunized and sacrificed at onset (*n* = 4 each), peak (*n* = 3 each), recovery (*n* = 3 each). CNS-infiltrating CD4+ T cells were isolated from these groups, as well as from mice that did not develop disease (*n* = 4 each) and the frequency of PD-1+CXCR5- cells was assessed. Data are from one of two immunizations. Tukey's post-hoc test after one-way ANOVA was used. **c** Expression of MHC class II (I–Ag7), IL-21, ICOS and CXCL13 were assessed from CNS-infiltrating CD4+ T cells taken from immunized male WT (*n* = 5) and IgH[MOG] (*n* = 5) mice, and expression between PD-1- and PD-1+ CD4+ subpopulations was compared. Data are representative of two immunizations. Representative plots are presented in Supplementary Fig. 5c. Sidak's multiple comparisons test after repeated measures two-way ANOVA was used. **d** Expression of IFNγ and IL-17 was assessed from PD-1- and PD-1+ CD4+ T cells

from immunized male WT or IgH[MOG] mice (*n* = 3 each). Data are representative of two immunizations. Sidak's multiple comparisons test after repeated measures two-way ANOVA was used. **e** Male WT (*n* = 5) and IgH[MOG] (*n* = 5) mice were immunized with MOG[35-55] and T cells were isolated from the dura mater and subdural meninges (SDM) at endpoints. Expression of PD-1 and CXCR5 was assessed by flow cytometry. Data are representative of two experiments for dura mater and one immunization for SDM. Two-tailed *t*-test was used. **f** T cells were isolated from spleen and CNS at endpoints from anti-p19 (*n* = 5) or isotype-treated (*n* = 4) immunized male IgH[MOG] mice. Expression of PD-1 and CXCR5 was assessed by flow cytometry. Data are representative of three experiments. Two-tailed *t*-test was used. In all violin plots, horizontal lines represent medians. Each datapoint represents an individual mouse. Datapoints connected by a line represent paired observations from the same mouse. Exact *p*-values can be found in the Source Data file. Linked to Supplementary Figs. 5, 6.

genotypes; IL-21 expression was negligible in splenic CD4+ T cells overall (Supplementary Fig. 5e). These findings indicate that PD-1+CXCR5- T cells express multiple known markers of Tph cells and are enriched in the CNS of immunized IgH[MOG] mice.

As we had found that the frequency of CNS-infiltrating IL-17+ CD4+ T cells is increased in immunized IgH[MOG] mice (Fig. 3c), we next asked whether production of IL-17 differed between the WT and IgH[MOG] Tph populations. Indeed, CNS-infiltrating IgH[MOG] CD4+PD-1+ T cells showed strikingly higher positivity for IL-17 than their WT counterparts (Fig. 5d). By contrast, we saw no differences in IL-17 between WT and IgH[MOG] CD4+PD-1- T cells, nor were strain-specific differences in IFNγ seen in either the CD4+PD-1+ or CD4+PD-1+ subpopulations. Thus, Tph cells are the primary source of IL-17 from the CD4+ T cell compartment in the IgH[MOG] CNS.

Human Tph cells share a number of transcriptional features with Tfh cells[27,65]. To ascertain whether this was the case for WT and IgH[MOG] CD4+PD-1+CXCR5- T cells, we sequenced the transcriptomes of these populations and compared them to the publicly available transcriptomes of murine Tfh and non-Tfh cells[66]. We found that both the WT and IgH[MOG] Tph-like transcriptomes bore greater similarity to Tfh cells than to non-Tfh cells, as measured by Euclidean distance (Supplementary Fig. 6a). Next, we investigated transcriptional differences between CNS-infiltrating WT and IgH[MOG] Tph-like cells. We uncovered 2565 differentially expressed genes ($p_{adj} < 0.05$ & $L_2FC > |1.5|$) of which 2154 were upregulated and 411 downregulated in IgH[MOG] versus WT (Supplementary Fig. 6b). Notably, and mirroring our observations at the protein level, expression of *Cxcl13*, *Il17a* and *Il17f* were increased in IgH[MOG] Tph cells (Supplementary Fig. 6c).

Tph cells promote the effector differentiation of B cells[27,65]. We therefore asked whether the PD1+CXCR5- T cells in our system could provide help to B cells in vitro. In a coculture system consisting of WT B cells with either PD1+CXCR5- or PD1neg T cells from WT or IgH[MOG] mice, we found that PD1+CXCR5- cells of both genotypes were better able to induce an IgMloIgDlo CS phenotype than their PD1neg counterparts (Supplementary Fig. 6d). Next, having seen an increase of B cell aggregates in the meningeal TLOs, we assessed by flow cytometry the presence of Tph cells in meningeal-associated tissue. An increase in the frequency of Tph cells was observed in the IgH[MOG] dura mater of IgH[MOG] mice compared to WT mice; we observed similar findings in the SDM, though the overall frequency of Tph cells was lower than in the dura mater (Fig. 5e).

Finally, we examined whether in vivo depletion of IL-23 could impact Tph frequency. Indeed, the frequency of Tph was reduced in the CNS, but not spleen, of immunized IgH[MOG] treated with anti-p19 (Fig. 5f). Altogether, our data show that CD4+PD1+CXCR5- Tph-like cells accumulate in the CNS of immunized IgH[MOG] mice in an IL-23-dependent manner, where they are an important source of IL-17 and CXCL13.

## Reactive oxygen species (ROS) generation is enhanced from IgH[MOG] B cells and Tph cells

We next sought to identify molecular features, characteristic of IgH[MOG] B cells, that might underpin the exacerbated disease seen in these mice. Pathway analysis of the IgH[MOG] versus WT CNS-infiltrating B cell transcriptome revealed, unsurprisingly, a number of gene ontology (GO) terms linked to leukocyte function and immunity. Strikingly, however, we also observed an over-representation of terms linked to metabolic processes, cellular respiration and neuronal death (Fig. 6a). Notably, similar GO terms were found to be upregulated when we compared the transcriptome of IgH[MOG] CNS Tph cells to WT (Fig. 6a); Indeed, *"pathways of neurodegeneration"* and *"oxidative phosphorylation"* gene sets were enriched in both IgH[MOG] B cells and Tph cells in the CNS (Fig. 6b). A cluster of 7 differentially expressed genes (DEG) in B cells shared both terms (Supplementary Fig. 7a). Interestingly, they included genes, such as *Lcn2* and *Casp3*, that are linked to neurotoxicity[67,68] as well as others, such as *Ralbp1* and *Apoe*, that are involved in dampening Alzheimer's disease (AD)-related mitochondrial dysfunction or degradation of amyloid-beta[69,70]. In Tph cells, 19 such common genes were identified (Supplementary Fig. 7b). They had roles in the neuroprotective heat-shock response, such as *Hsf1*[71] and *BagS*[72]; autophagy, such as *Pink1*[73] and *Atg7*[74]; or, like *Ptk2b*[75] and *App*[75], were linked to AD pathology.

These findings suggested to us that an oxidative stress program might underly the EAE phenotype of IgH[MOG] mice. We therefore examined whether CNS-infiltrating B cells and Tph cells could produce reactive oxygen species (ROS). Notably, IgH[MOG] B cells (Fig. 6c) and Tph cells (Fig. 6d) had higher ROS production than their WT counterparts, as assessed by incorporation of CM-H2DCFDA, a fluorescent oxidative stress indicator. Then, we examined whether in vivo depletion of IL-23 could impact ROS production from IgH[MOG] B and Tph cells. Interestingly, IL-23 blockade decreased ROS generation from CNS-infiltrating B cells (Fig. 6e) and Tph cells (Fig. 6f). In sum, these data show that CNS-infiltrating IgH[MOG] B cells and Tph cells mount oxidative stress responses that are IL-23 dependent.

## Discussion

IgH[MOG] mice were initially described on the C57BL/6 (B6) and SJL/J genetic backgrounds. On the B6 background, IgH[MOG] mice showed an increased incidence of EAE, relative to WT, when immunized with whole MOG protein[17]. Notably, IgH[MOG] SJL/J mice developed EAE of greater severity than controls when immunized with the myelin-derived epitope proteolipid protein (PLP)[139-154][17], which induces a relapsing/remitting disease pattern in SJL/J background mice[76]. This suggested that the presence of MOG-reactive B cells could contribute to EAE pathology that was driven by a class II-restricted peptide. It was later shown that when IgH[MOG] mice were crossed to the 2D2 MOG[35-55] TcR-transgenic strain on the C57BL/6 background, the resulting

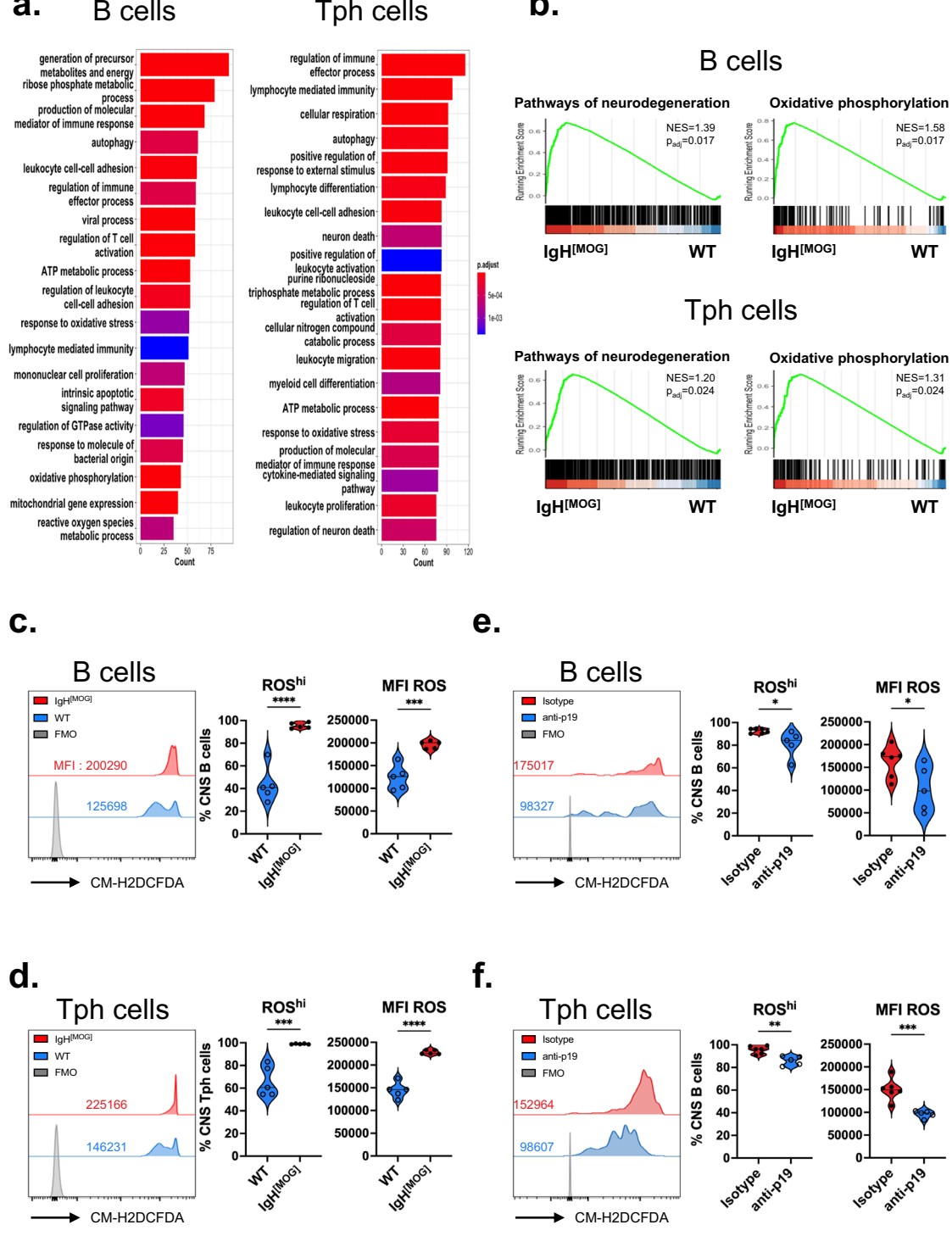

**Fig. 6 | CNS-infiltrating IgH[MOG] B cells and Tph cells show increased production of reactive oxygen species. a** GO term analysis of the transcriptome of CNS-infiltrating B cells and Tph cells from female IgH[MOG] (*n* = 3) vs WT (*n* = 3) from a single immunization. **b** Normalized enrichment scores (NES) for "pathways of neurodegeneration" (mmu05022) and "oxidative phosphorylation" (mmu00190) from B cell and Tph cell transcriptomes from the same samples as in (**a**). The weighted Kolmogorov–Smirnov statistical test with the Benjamini-Hochberg method was used to adjust for the false discovery rate. Uptake of ROS indicator CM-H2DCFDA from B cells (**c**, **e**) or Tph cells (**d**, **f**) from immunized female IgH[MOG] vs WT mice (**c**, **d**; *n* = 5 each) or from immunized and anti-p19 (*n* = 5) vs isotype-treated (*n* = 6) immunized female IgH[MOG] mice. Data in **c**, **d** are representative of two immunizations. Data in **e**, **f** are representative of two immunizations. Two-tailed *t*-test was used. In all violin plots, horizontal lines represent medians. Each datapoint represents an individual mouse. Exact *p*-values can be found in the Source Data file. Linked to Supplementary Fig. 7.

double-transgenics spontaneously developed Devic's-like disease while single transgenic mice did not[18]. Similarly, IgH[MOG] x TCR[1640] SJL/J[19] and IgH[MOG] x 1C6 NOD[31] double-transgenic mice develop spontaneous EAE at a high rate of incidence[31]. These findings indicated that the collaboration of both myelin-specific B and T cells in the same animal could induce CNS autoimmunity; however, it was difficult to determine whether B or T cell-driven responses were initially responsible for disease induction in this model, and thus the nature of a

putative collaboration between B cells and T cells in disease processes remained incompletely defined.

In our study, we actively immunized IgH[MOG] mice on the NOD background with the class II-restricted peptide MOG[35-55]. This permitted us to study the contribution of MOG-reactive B cells in a model of EAE that is initiated by CD4+ T cells. While WT NOD mice developed a gradual, chronic MOG[35-55]-driven disease course, with advanced symptoms appearing as late as >100 days post immunization, we found that immunized IgH[MOG] NOD mice develop severe disease within a matter of weeks that is characterized by the accumulation of meningeal lymphocytic aggregates reminiscent of TLOs. Thus, by using a myelin-derived, class II-restricted, immunogen, we show that B cells augment EAE even when CD4+ T cells initiate disease. Further, MOG[35-55]-immunized IgH[MOG] mice represent an attractive model by which to dissect the role of TLOs in the subpial damage and neurodegeneration that is seen in progressive MS.

Both male and female IgH[MOG] mice mounted severe disease, indicating that the phenotype is not sex-regulated. Indeed, actively immunized NOD mice show no sex difference with respect to EAE[77]. However, we recently showed that encephalitogenic Th17 responses on the NOD background are skewed towards males[33]. Thus, it is possible that there exists a subtle role for biological sex in regulating B:Th17 cell interactions in IgH[MOG] mice; this may merit further investigation.

The role of CD4+ T cells in licensing peripheral B cell responses is well-established. Th17 cells, in particular, can foster B cell responses by inducing class switching to inflammatory IgG1 and IgG2a[78] and by recruiting B cells to TLO structures[79]. Here, we provide evidence that the converse may also be true; CNS-infiltrating CD4+ T cells in IgH[MOG] mice showed significantly higher expression of IL-17 and GM-CSF, but not of IFNγ, indicating that Th17 responses are elevated in these mice. Further, CNS-infiltrating IgH[MOG] B cells showed increased expression of IL-23, an innate immune-associated cytokine that stabilizes the Th17 lineage and is required for the pathogenicity of Th17 cells[53,54], and in vivo blockade of IL-23 sharply reduced EAE in IgH[MOG] mice. Importantly, there is emerging evidence that B cells can promote Th17 responses. Both peritoneal CD5+B220lo B cells[21] and LPS-stimulated splenic CD80+CD86+CD44+ B cells[22] can elicit Th17 differentiation. Further, production of IL-17 is defective from T cells taken from agammaglobulinemia patients that lack B cells[23] and therapeutic depletion of B cells with rituximab can reduce Th17 function in the context of MS[24] as well as in rheumatoid arthritis[25]. The mechanisms underpinning bidirectional B: T cell interactions remain to be fully elucidated.

Human B cells can secrete IL-23, and repression of this capacity is posited as a mechanism of action for interferon-beta in MS[55]. It was previously observed that peripheral B cells from naïve IgH[MOG] mice possess the capacity for IL-23 secretion[80]; both upon pulsing with rMOG, and most strikingly upon co-culture with MOG[35-55]-specific 2D2 T cells in the presence of either rMOG or MOG[35-55]. B cell production of IL-23 in response to LPS is repressed by the Tim-1 inhibitory receptor[81]. We now show that immunization with MOG[35-55] elicits IL-23 expression from CNS-infiltrating B cells in vivo, and that IL-23 is essential to both EAE and TLO formation from these animals.

Using mass cytometry, Brenner and colleagues found that PD1+CXCR5-CD4+ T cells accumulate in the inflamed synovium of RA patients. These Tph cells secrete B cell attractant factors such as CXCL13 and IL-21 and associate with B cells in the inflamed synovium, both in defined lymphoid aggregates as well as more diffusely. However, Tph are not defined by expression of Bcl6, the Tfh master transcription factor[27]. While these cells are also observed in patients with lupus[62] and ANCA-associated vasculitis[82], their potential roles in murine models of autoimmunity remain obscure. We have found an enrichment of PD1+CXCR5- CD4+ T cells in the CNS, but not immune periphery, of immunized IgH[MOG] mice. These cells are Bcl6neg, show

increased expression of the Tph markers MHC class II, ICOS, CXCL13 and IL-21 relative to PD-1- counterparts, and elicit B cell class-switching in vitro. Intriguingly, we find that the presence of PD-1+CXCR5- cells is highest in mice of peak EAE severity; Tph cells were previously shown to correlate with the severity of rheumatoid arthritis[27] and lupus[62]. Here, we also show that Tph cells can be found in the dura mater of immunized mice and their frequency was higher in IgH[MOG] mice. Further, the accumulation of Tph cells in the CNS is dependent on IL-23, revealing a heretofore unknown function of this cytokine.

CXCL13 production characterizes human Tfh[83] and Tph[27] cells. Despite this, our finding that IgH[MOG] Tph-like cells express CXCL13 was curious, as it has been proposed that murine Tfh cells do not generate this chemokine[84]. Notably, however, several recent lines of evidence indicate that murine CD4+ T cells are capable of CXCL13 production[63,64]. Our findings thus add to an emerging literature suggesting that CXCL13 expression may characterize specific subsets of murine T cells under certain conditions; the functional significance of these observations requires further elucidation.

Oxidative stress is an attractive mechanistic explanation for the CNS damage observed in MS. Indeed, markers of oxidative stress are increased in the CSF of MS-affected individuals[85] and oral dimethyl fumarate, which invokes a cellular antioxidant response, is a well-established treatment for MS[86]. Oxidative stress may be responsible for the subpial cortical demyelination that is characteristic of progressive MS, though this has been thought to be caused primarily by inflammatory macrophages and microglia[87]. Here, we find evidence that oxidative phosphorylation and neurodegeneration pathways, as well as ROS production, are upregulated in CNS-infiltrating IgH[MOG] B and Tph cells; suggesting that lymphocytes may also be a key contributor to oxidative injury in progressive disease. Intriguingly, we show that ROS production from these cells is IL-23-dependent; further work is needed to grasp the underlying molecular cues.

In conclusion, we provide evidence that in the presence of a B cell repertoire that is skewed towards MOG, NOD-background mice develop unusually rapid and severe CD4+ T cell-mediated EAE. A robust frequency of IgH[MOG] B cells unexpectedly produce IL-23; severe disease and meningeal lymphocytic accumulation in these animals is IL-23-dependent. Finally, Tph cells aggregate in the CNS and the dura mater of these mice. Together, our findings support a critical role for myelin-reactive B cells in bolstering T cell-driven CNS autoimmunity in an IL-23-dependent manner.

## Methods

### Ethics
All mouse experiments and breedings were approved by the Animal Protection Committee of Université Laval (protocols 2021-820 and 2021-830, to M.R). Protocols and experiments involving human participants were approved by the Newfoundland Health Research Ethics Board (to C.S.M).

### Animals
NOD/ShiLtJ (WT) and NOD.Scid mice were purchased from Jackson Laboratories. IgH[MOG] mice on the NOD background[31], and 1C6 mice[31-33], were obtained from Dr. Vijay Kuchroo (Brigham & Women's Hospital, Boston), and were maintained at the animal facility of the Centre de recherche du CHU de Québec-Université Laval, with a 12 h/ 12 h light cycle, a temperature maintained at 23 °C ± 2 °C and relative humidity at 50% ± 5%. Mice were used between 9 to 12 weeks of age. The sex of mice used in each experiment have been described in the figure legends.

### EAE induction and scoring
WT NOD and IgH[MOG] mice were immunized subcutaneously with 200 µg MOG[35-55] (synthesized at the CHU de Québec), emulsified in incomplete Freund's adjuvant (BD Difco) that was supplemented with

500 µg *M. tuberculosis* extract (BD Difco). On day 0 and day 2 post-immunization, mice received 200 ng pertussis toxin (List Biological Laboratories) intraperitoneally. For p19 blockade, mice were administered anti-p19 (clone G23-8) or isotype (rIgG1, clone HRPM; both BioXcell), 1 mg, on d[-1] and d[+6] post-immunization[56]. For the comparison of Th17 cell pathogenicity in the presence of WT or IgH[MOG] B cells (Fig. 3d), 10-week-old NOD.*Scid* mice were first passively infused i.v. with $2 \times 10^6$ CD19[+] WT or IgH[MOG] B cells from unimmunized donors. Fourteen days post-infusion, $2 \times 10^6$ in vitro-differentiated 1C6 Th17 cells were introduced i.v. to the same recipient animals, similar to as we previously described[33]. Mice received 200 ng pertussis toxin on d0 and d2 and were monitored daily, as above. All EAE mice were scored daily for signs of disease, which were assessed using a semi-quantitative 0-5 scale: 0; no disease, 0.5; ruffled fur, 1; limp tail, 1.5; mild impairment in gait, 2; severe impairment in gait, 2.5; partial hind limb paralysis, 3; hind limb paralysis, 4; forelimb paralysis, 5; ethical endpoints attained[88]. Mouse body condition was also monitored daily by trained veterinary staff, in consultation with the Université Laval veterinary service, as required by our animal use protocol. In Fig. 1, immunized IgH[MOG] and WT mice were followed for 25 days as indicated. In subsequent active immunization experiments, IgH[MOG] mice were considered to be at endpoint when they either reached ethical endpoints as determined by the Université Laval veterinary service, or when they held a disease score of ≥ 3 for 3 consecutive days without remission. In studies requiring ex vivo analyses, WT comparator mice were sacrificed and analyzed in parallel as IgH[MOG] mice attained endpoints. In the adoptive transfer experiments, recipient mice were followed for up to 70 days or until ethical endpoints were attained. For comparison of disease burden, AUC[33,89] was first calculated for individual disease curves and then statistical comparisons were made between individual groups. For AUC comparisons, mice attaining ethical endpoints were assigned a score of 5 from that point until the end of the experimental monitoring period. For comparison of rapidity of disease worsening, linear regression was conducted on pooled disease curves from individual experiments[33,88,89].

## Histopathology

Mice were euthanized and intracardially perfused with PBS. The spinal cord was extracted and fixed in 10% formalin and 30% sucrose prior to being embedded in paraffin. Five-to-seven-micron coronal sections of the spinal cords were collected on superfrost microscope slides (using microtome, Leica Biosystems HistoCore Autocut) and dried overnight on a drying bench at 37 °C. Slides were stored at room temperature in a slide storage box until staining was performed. Sections for Hematoxylin and Eosin (H&E) were deparaffinated with xylene, while toluene was used for Luxol Fast Blue (LFB) sections. Histology was performed as previously described[90], using standard H&E to visualize lymphocyte infiltration and LFB to stain myelin in order to visualize areas of demyelination. Briefly, stained slides were scanned at 20x using a brightfield microscope scanner (Aperio AT2 DX System), and representative RGB images of the cervical spine were acquired. Quantification of the staining was performed on Image J (software, v 1.53 K) in a blinded manner on three sections per sample (one cervical, one thoracic, one lumbar) for spinal cord, or on 3 separate regions of interest for brain samples. Images were split into separate single color channels using the Color Deconvolution plugin. Thresholding was performed on the single color channels for hematoxylin and LFB, and an area fraction measurement was performed on the white matter of the spinal cord. Staining is expressed as the percent of total white matter that is stained.

## Immunofluorescence (IF)

Heads of euthanized mice were decalcified in EDTA for up to 2 weeks and were subsequently embedded in optimum cutting temperature medium (OCT). The tissue was cryo-sectioned (7-10um) and mounted onto positively charged slides that were subsequently stained with IF antibodies. Slides were imaged using a widefield inverted microscope (AxioObserver 7) and analysis was done using Image J software (v 1.53 K). $5 \times 5$ stitched scans of the whole brain, along with 20x and 40x images of meningeal lymphocyte clusters and brain parenchyma were captured. Counting of meningeal lymphocyte clusters was performed using $5 \times 5$ stitched images that encompassed the entire brain. The meningeal compartment was discerned within these images, and TLOs were recognized as clusters of B220[+] B-cells within the meninges. The presence of CD4[+] T-cells and CD11c[+] dendritic cells was verified within these clusters through 40x images. For IgG deposition, 20x images of three parenchymal regions per tissue were subject to background subtraction, following which the images were converted into grayscale and thresholding was performed. The average pixel gray value (mean gray value) and percent area stained with IgG were calculated in each image.

## Cytokine ELISA

Splenocytes and draining lymph nodes were isolated from NOD and IgH[MOG] mice 5d post-immunization. Cells were cultured at a concentration of $10 \times 10^6$ cells mL[-1] in T cell media and stimulated, or not, with 10 mg mL[-1] MOG[35-55]. Supernatants were collected at d5 of the culture for the analysis of T cell cytokine production using a combination of commercially available antibodies (Biolegend). Briefly, the capture antibodies used were: purified anti-mouse IFNg (clone R4-6A2) and purified anti-mouse IL-17 (clone TC11-18H10.1). The detection antibodies used were: biotin anti-mouse IFNg; clone XMG1.2, biotin anti-mouse IL-17; clone TC11-8H4. Following the incubation with the avidin-horseradish peroxidase (Biolegend) and TMB substrate (Mendel Scientific), colorimetric readings were performed using a SpectraMax i3 Microplate Reader.

## Measurement of murine serum immunoglobulin

Blood was collected from IgH[MOG] mice and WT mice at endpoint. Serum was collected by centrifuging blood samples at 2000*g* for 10 min. Total anti-MOG IgG was quantified by using SensoLyte Anti-Mouse MOG(1-125) IgG Quantitative ELISA Kit (Anaspec).

## Isolation of CNS parenchyma-infiltrating mononuclear cells

Mice were euthanized and perfused intracardially with PBS. Brain and spinal cord were dissected from the skull and vertebral column respectively and were prepared as previously described[88]. Briefly, CNS tissues were digested with liberase (Roche) and DNAse I (Sigma) and cells were enriched using a 35% Percoll (GE Healthcare) gradient.

## Isolation of dura mater and subdural-enriched meningeal cells

Mice were injected intravenously with fluorochrome-labeled anti-CD45 antibody (CD45iv), 15 min prior to euthanasia, to permit the exclusion of peripheral leukocytes in analyses[79]. Mice were then perfused intracardially with PBS and the skull was gently removed mechanically to peel the dura from the edges of the skull cap. Subdural enriched meninges were gently peeled off from cerebral hemispheres. Samples were then incubated for 30 min at 37 °C in digestion buffer: 1 mg mL[-1] collagenase D (Sigma) and 50 µg mL[-1] of DNAse 1 (Sigma) in RPMI. Samples were mechanically dissociated through a 100 µm cell strainer and flowthrough was centrifuged at 300 *g* for 5 min at 4 °C. Finally, the supernatant was discarded, and cell pellets were resuspended for flow cytometry analysis of live CD45[+]CD45iv[neg] cells.

## Flow cytometry

Single cell suspensions were obtained from spleens, lymph nodes (draining/inguinal or cervical) and CNS of EAE mice. For detection of surface antigens, cells were incubated with Fc Block (Biolegend) and stained with Fixable Viability Dye (VD; eBioscience) prior to staining

with antibodies against surface antigens (CD45, CD44, CD4, CD8, CD19, CD11b, CD11c, B220, Ly6G, IgD, IgM, FAS, GL7, PD-1, CXCR5, ICOS, I-Ag[7], GFAP; details in following section). For detection of intracellular cytokines, cells were first stimulated with 50 ng ml[-1] PMA (Sigma), 1 μM ionomycin (Sigma) and 1 μL mL[-1] GolgiStop (BD) for 4 h at 37 °C, prior to being labeled with viability indicator, Fc Block and relevant surface antigens as above. They were then fixed and permeabilized (Fixation Buffer and Intracellular Staining Perm Wash Buffer, both Biolegend) and stained for intracellular markers (IFN-γ, IL-17A, TNF, IL-6, GM-CSF, IL-12p35, IL-23; Bcl6, IL-21, CXCL13; details in following section). CM-H2DCFDA, a general oxidative stress indicator, was used to detect total ROS production. Cells were incubated with 1 μM CM-H2DCFDA (Thermo Fisher Scientific, C6827) at 37 °C for 30 min, followed by surface proteins staining as described above. Biotinylated mouse MOG protein (ACRO Biosystems, MOG-M82E7), followed by streptavidin-PerCP-Cyanine5.5 conjugate (Invitrogen) was used for MOG-specific B cell enumeration. Samples were analyzed on a FACS Aria (BD) and data were analyzed using FlowJo software (v 10.9.0) (Treestar). Positive/negative discrimination was set on the basis of fluorescence minus one (FMO) controls and all data were pre-gated on live cells as determined by viability indicator. A sample gating strategy is provided.

## Flow cytometry antibodies

The following monoclonal antibodies against mouse antigens were used: CD45, clone A20 (Biolegend); CD11b, clone M1/70 (eBioscience); CD11c, clone N418 (Biolegend); Ly6G, clone 1A8 (BD Biosciences); CD4, clone RM4-5 (eBioscience); CD8, clone 53-6.7 (Biolegend); CD44, clone IM7 (Biolegend); CD19, clone 1D3 (eBioscience); B220, clone RA3-6B2 (eBioscience); IgD, clone 11-26c (eBioscience); IgM, clone 11/41 (eBioscience); FAS, clone 15A7 (eBioscience); GL7, clone GL-7 (eBioscience); PD-1, clone J43 (eBioscience), CXCR5, clone SPRCL5 (eBioscience); ICOS, clone C398.4 A (eBioscience); GFAP, clone 1B4 (BD Biosciences); MHC class II (I-Ag[7]), clone 39-10-8 (Biolegend); IFN-γ, clone XMG1.2 (eBioscience); TNF, clone MP6-XT22 (eBioscience); IL-17a, clone TC11-18H10.1 (Biolegend); GM-CSF, clone MP1-22E9 (eBioscience); IL-6, clone MP5-20F3 (eBioscience); IL-21, clone mhalx21 (eBioscience); IL-23 heterodimer, clone IC18871P (R&D Systems); IL-23p19, clone fc23pg (eBioscience); IL-12p35, clone 4D10p35 (eBioscience); IL-12p40, clone C17.8 (eBioscience); BCL6, clone IG191E/A8 (eBioscience); CXCL13, clone D58CX13 (eBioscience); CM-H2DCFDA, clone C6827 (Thermo Fisher Scientific).

## Immunofluorescence antibodies

The following monoclonal antibodies against mouse antigens were used: CD4 (RM4-5), B220 (RA3-6B2), CD11c (N418), CD45.2 (104), PDGFRα (APA5), PDGFRβ (APB5), CXCL13 (PA5-47018) and AID (13-5959-80), all of which are from eBioscience; fibronectin (GW20021F, Sigma); smooth muscle actin (1A4, Sigma); IL-23 heterodimer (IC18871P, R&D Systems); CD138 (142504, Biolegend); IgG (1144-02, SouthernBiotech).

## T cell antigen presentation assay

Single cell suspensions were obtained from the spleens of unimmunized WT and IgH[MOG] mice. Cells were labeled with CD43 (Ly-48) Microbeads (Miltenyi), and CD43[+] leukocytes (all leukocytes except resting B cells) were depleted on a magnetic MACS column (Miltyeni). Unlabeled CD43[-] cells were collected and were subsequently stained with anti-mouse CD19. CD19[+] B cells were purified using high-speed cell sorting. In parallel, CD4[+] T cells were purified from the spleens of 1C6 mice using mouse CD4 MicroBeads (Miltenyi) and labeled with CellTrace Violet (CTV; Thermo Fisher Scientific). CTV-labeled 1C6 CD4[+] T cells were cultured with B cells at a ratio of 1 CD4: 1 B, with 0, 1 or 10 μg ml[-1] MOG[35-55] for 72 h. CTV dilution was assessed by flow cytometry.

## Tfh culture

The protocol of Gao et al. was adapted[91]. Briefly, 5 × 10[5] splenocytes from WT mice were stimulated 24 h with LPS (1 μg mL[-1]). 1 × 10[4] MOG[35-55]-specific 1C6 CD4[+]CD62L[hi] T cells were then plated on top of the splenocytes with 1 μg mL[-1] MOG[35-55], plus 50 ng ml[-1] IL-21 and 100 ng mL[-1] of IL-6 (Tfh) or no additional cytokines (Th0) for 72 h.

## Tph-B cell help assay

Modifying a previously described protocol[91], 2 × 10[5] B220[+]CD19[+] splenic B cells from naïve NOD mice were purified and pre-stimulated with LPS (1 μg mL[-1]) for 24 h. Next, they were co-cultured at a 1:1 ratio with purified splenic CD4[+]CXCR5[-]PD-1[+] or CD4[+]PD-1[neg] T cells from either WT or IgH[MOG] immunized mice. After 6 days of co-culture, live B220[+] cells were assessed for their expression of IgM and IgD by flow cytometry.

## Analysis of CSF from MS-affected individuals

5 mL of CSF was collected from consenting adult participants with MS at The Health Sciences Centre neurology clinic in St. John's, NL between July 2021 and May 2022. For CSF immune phenotyping, 2.5 mL of CSF was centrifuged for 10 min at 300 g and the CSF was removed. The CSF cell pellet was resuspended in 100 uL of flow buffer (1% bovine albumin serum, 2 mM EDTA, 2 mM sodium azide in PBS). The CSF cell suspension was added to a DURAclone IM Phenotyping BASIC tube (Beckman Coulter), mixed and incubated at 4 °C for 30 min. The cells were washed with 4 mL flow buffer, centrifuged at 300 g for 5 min, decanted and resuspended in 100 μL 2% paraformaldehyde. Data was acquired from the whole sample using a Cytoflex flow cytometer (Beckman Coulter) and analyzed using FlowJo software. The CD19[+] cell count was expressed as a percentage of the total number of CD45[+] cells.

For measurement of CSF cell IL23A expression, 2.5 mL of human CSF was centrifuged at 300 g for 10 min and the CSF supernatant was carefully removed. The CSF cell pellet was resuspended in 500 ul of Trizol and stored at − 80 °C. Total RNA was extracted using chloroform extraction followed by an RNeasy Micro kit (Qiagen). 200 ng of RNA was used to synthesize cDNA with the M-MLV Reverse Transcriptase kit (Invitrogen). IL23A transcript expression was quantified using IL23A (FAM labeled) and GAPDH (VIC labeled) Taqman probe/primer assays (Invitrogen), Fast Advanced Master mix (Invitrogen) and a ViiA7 Real-time PCR system (Applied Biosystems).

## mRNA-seq

At experimental endpoints, CNS tissues from WT and IgH[MOG] mice were necropsied and mononuclear cells were isolated. Live CD19[+] B cells and CD4[+]PD1[+]CXCR5[-] Tph cells were purified by high-speed sorting (FACSAria II) and were directly processed using NEBNext Single Cell/Low Input RNA Library Prep Kit (New England Biolabs), according to the manufacturer's instructions, for mRNA sequencing library preparation. Total mRNA was enriched using poly(A)+ primers and template-switching oligo synthesize double-stranded cDNA with reverse transcriptase. A PCR amplification of 21 cycles was performed to increase cDNA yields followed by a purification step with AxyPrep Mag PCR Clean-up kit (Axygen, Big Flats, NY, USA). Enzymatic fragmentation of cDNA and end-repair steps were followed by ligation of NEBNext adapters and PCR enrichment step of 11 cycles to incorporate specific indexed adapters for the multiplexing. The quality of final amplified libraries was examined with a DNA screentape D1000 on a TapeStation 2200 and the quantification was done on the QuBit 3.0 fluorometer (ThermoFisher Scientific, Canada). Subsequently, mRNA-seq libraries with unique dual indexes were pooled together in equimolar ratio and sequenced for paired-end 100 pb sequencing on NovaSeq 6000 at the Next-Generation Sequencing Platform, Genomics Center, CHU de Québec-Université Laval Research Center. The

average insert size for the libraries was 253 bp. The mean coverage/sample was 24 M paired-end reads.

## Bioinformatics analyses

Paired-end mRNAseq fastq datasets were trimmed with trimmomatic v0.39 using the following command:

*java -jar trimmomatic-0.39.jar PE -phred 33 sample_R1.fastq sample_R2.fastq trimmed/paired_sample_R1.fastq trimmed/unpaired_sample_R1.fastq trimmed/paired_sample_R2.fastq trimmed/unpaired_sample_R2.fastq ILLUMINACLIP:TruSeq3-PE.fa:2:30:10 LEADING:30 TRAILING:30 SLIDINGWINDOW:4:15 MINLEN:30.*

Trimmed fastq files were mapped on the *M. musculus* or *H. sapiens* genome (Ensemble; Mus_musculus.GRCm39.cds.all.fa.idx; Homo_sapiens.GRCh38.cds.all.fa.idx) and quantified using kallisto 0.44.0. Prior to differential gene analysis, low count reads (mean of < 10 counts across the two genotypes) were filtered out[92]. Differential expression between WT and IgH[MOG] cells was performed using DESeq2 v1.38.3. For identifying relevant pathways, we used Gene Ontology enrichment analysis to focus on biological processes. We also used the simplify function from ClusterProfiler package to remove redundant terms. Finally, GSEA from the Broad Institute of UCSD was used to calculate the enrichment score of different gene sets. For Euclidean distance analysis, we used Tfh and non-Tfh samples (SAMN20569976-SAMN20569978 & SAMN20569991-SAMN20569993) from a publicly available transcriptomic dataset (Bioproject PRJNA752042). Non-Tfh was referred to as Tconv in the initial study[66]. The centroid of each group (Tfh and non-Tfh) was determined and the Euclidian distance between these and each PD-1+CXCR5- or PD1-CXCR5- transcriptome was then calculated using the function "euc.dist <- function(x1, x2) sqrt(sum((x1 - x2) ^ 2))" that was applied for each samples. For the *IL23A* analysis in MS B cells, we used all BioSamples from a publicly available transcriptomic dataset (Bioproject PRJNA727413).

## Statistics & reproducibility

Group sizes for active immunization EAE were determined by power analysis. Group sizes for ex vivo and histopathological analyses were based on previous literature[89,90,93]. No data were excluded from the analyses. Mice were distributed randomly into biological groups. The investigators were not blinded to allocation during experiments and outcome assessment. Comparisons between two groups were made by *t*-test. Fisher's exact test was used to test for differences in the frequency of mice attaining ethical endpoints. Comparisons between more than two groups were made by ANOVA, with the specific post-hoc test indicated in each Figure legend. Linear regression analysis was used for some EAE experiments and for human CSF analysis; Spearman r was calculated for the latter. For bioinformatics analyses, $p_{adj}$ is reported (weighted Kolmogorov–Smirnov statistical test with the Benjamini-Hochberg method used to adjust for the false discovery rate). Figure legends indicate cases in which data represent paired observations. Two-tailed analyses were used in all instances. In violin plots, horizontal lines represent the medians. In bar graphs, error bars represent s.e.m. Statistical analyses were conducted using Prism software v. 10.0.3 (GraphPad), or RStudio v2023.03.1 + 446 (Posit) in the case of bioinformatics.

## Reporting summary

Further information on research design is available in the Nature Portfolio Reporting Summary linked to this article.

## Data availability

The original transcriptomic data used in this publication have been deposited in NCBI's Gene Expression Omnibus (GEO) platform under accession code GSE267422 and BioProject PRJNA1021552. Publicly available transcriptomic data used for Supplementary Fig. 4g is available under accession code GSE173789 and for Supplementary Fig. 6a under accession code GSE181433. Source data are provided with this paper.

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

## Acknowledgements

We thank Vijay Kuchroo for providing us with NOD-background IgH[MOG] and 1C6 mice respectively. We thank Françoise Morin for critical discussions; Vincent Desrosiers, Alexandre Brunet, Stéphanie Fiola for technical assistance with flow cytometry; and Kim Larose, Andrée Brisson, Mathieu Vallière-St-Amant, Cindy Ouellet and the veterinary service of Université Laval for technical assistance and collaboration regarding animal care. The work was supported by a Biomedical Discovery Research Grant from the MS Society of Canada #3781 (M.R.), a CIHR Project Grant #PJT-178109-2021 (M.R.) and a UHN-Krembil startup fund #410014711(O.R.). M.R.F. and P.G. are supported by Ph.D. studentships from the Fonds recherche du Québec – Santé (FRQS). PMIAD was supported by a doctoral studentship from the Multiple Sclerosis Society of Canada. N.B. is Junior-2 scholar, and M.R. is a Senior scholar, of the FRQS.

## Author contributions

M.R.F. directed the project conducted experiments and helped write the manuscript. P.M.I.A.D., N.F., I.A., A.P.Y. and J.B. conducted experiments. R.P., A.R., B.M., S.L. and O.R. conducted histological analyses. P.G. and N.B. assisted with serum detection of immunoglobulins. C.J.-B. and A.D. assisted with bioinformatics analyses. C.S.M. contributed to writing of the manuscript. M.R. supervised the project and wrote the manuscript. All authors read and approved the final manuscript.

## Competing interests

M.R. held a research contract with Remedy Pharmaceuticals (2019-2021), with funds paid to the CHU de Québec and has conducted educational activities for Novartis Canada. These activities are unrelated to the work in this manuscript. The remaining authors declare no competing interests.
