## [Peer Review File · Nature Communications]

Myelin-reactive B cells exacerbate CD4⁺ T cell-driven CNS autoimmunity in an IL-23-dependent mannerREVIEWER COMMENTS

Reviewer #1 (Remarks to the Author):

In the last several years, B cells have been found to play important roles in MS pathogenesis in addition to T cells, which are largely considered to be the drivers of MS pathophysiology. The success of B cell-depleting therapies has demonstrated their clinical relevance in MS but the biological mechanisms are only partially understood, while many 'T cell' therapies also show an impact on B cells. In this paper, Fazazi et al. investigate the mechanisms behind how MOG reactive B cells mediate T cell driven pathogenicity, which has not been fully elucidated before. To do this, the authors induce MOG35-55 active EAE in IgH[MOG] mice, in which the B cells are skewed to recognizing the MOG protein. Using this model, they show that the presence of MOG reactive B cells further exacerbates EAE clinical scores during active EAE without altering peripheral T or B cell activation or MOG specific IgG production. In contrast, in the CNS, they show increased accumulation of tertiary lymphoid organs in the meninges and increased cytokine production by the B cells and Th17 cells of the IgH[MOG] mice. They also show that adoptive transfer of IgH[MOG] B cells in addition to Th17 cells results in exacerbation of EAE clinical scores in the passive EAE model. IL23, a Th17 stabilization factor, has been previously shown to be secreted by B cells, and in their model only B cells (not DCs, astros or microglia) from IgH-MOG secrete higher levels of IL-23 compared to WT. The authors further show that blockade of IL23p19 using a blocking antibody resulted in amelioration of EAE and reduced the presence of TLOs and frequency of T peripheral helper cells in the CNS of the treated mice. They also show a correlation between the proportion of B cells in the CSF of pwMS and levels of IL23 mRNA, suggesting that this mechanism could be at play in MS as well. They finally determined that pathways associated with neurodegeneration and oxidative stress are overrepresented in the IgH[MOG] B cells and T peripheral helper cells compared to wild type, before showing that blockade of IL23 reduces reactive oxygen species production by B cells and Tph cells.

Overall, using multiple EAE models and approaches, the authors convincingly show that B cells facilitate pathogenic Th17 function, suggesting that B cell-Th17 cell interactions are essential in CNS autoimmunity and at least partially dependent on IL-23. The methodology is sound and the methods section is sufficiently detailed. The conclusions made are supported by the evidence shown and the mechanism identified is both relevant (scientifically and clinically) and significant to the field.

Minor concerns to address:

-Demonstrating that B cells from MS patients in either peripheral blood or CSF produce higher levels of IL-23 by ELISA or express IL-23 more frequently by FACS or mRNA for il23a, generating new data or mining published scRNAseq or bulk RNAseq from B cells of MS vs. healthy controls, or showing B cells expressing IL-23 in the CNS of MS patients, or showing that B cells from MS express lower levels of IL-23 after induction therapy or B cell-depletion would be an interesting addition to

emphasize the importance of the mechanism identified in human MS.

-In Figure 6, it would be more informative to see some of the specific genes associated with oxidative stress and neurodegeneration that are differentially expressed in the IgH[MOG] mice, either as a table of the fold changes and p-values or graphs of the normalized counts.

-‘Disease is dependent on IL-23, which is produced by B cells and is required for the augmented Th17 responses seen in these mice.’ As the authors have shown that other immune cells (DCs, monos, macros) also produce IL-23, although there are no differences between WT and IgH-MOG in terms of IL-23 production by these subsets contrarily to what they found in B cells, this statement should be reformulated.

In general the number of animals, sections, replicates could be more clearly indicated (indicate if n refers to the number of animals, of sections, etc.). Show all points clearly (not overlapped) and describe what the lines are indicating. For example:

-Figure 1b: figure legend describes n of 4 but there are only two points for some? please describe the number of animals and the number of sections per animal studied for each region, and ensure that $n \geq 4$.

-Figure 1d: representative of triplicates from how many animals?

-Figure 2b: why are there two points next to each other X 3 for IgH-MOG mice (quantification)?

-Figure 4b: please do not use two different shades of blue for two different markers, CD4 expression is difficult to discern. In higher magnification image the cells expressing B220 seem to lack nuclei (DAPI). Please provide new images.

-There are multiple places in the manuscript where the reference superscript is preceded by “(ref)”, please fix this typo

-Line 178 a/b typo following PDGFR

-Fig. S4C mislabeling?

-The font sizes of the axes for many of the graphs are very small and should be increased to help with visibility.

-Is there a reason why the first graphs in figure 1 with the EAE curves are split by sex and then this is not done for later analyses? Can you specify this reason in the results and/or discussion and briefly discuss how sex of the mice could or could not play a role in the mechanisms you identified? And what this would imply for MS pathogenesis, where sex is a risk factor depending on the type?

-In Figure 6 c-f, combining the graphs of the ROS production in WT & IgH[MOG] and the Isotype and a-IL23 treated mice for each cell type would be more visually striking. If there is a reason to keep

them separate, please explain.

-The oxidative stress paragraph in the discussion is minimal, please expand on what is known about oxidative stress in B cells and Tph cells and how this ties into your data and MS pathology.

Reviewer #2 (Remarks to the Author):

Fazazi and colleagues provide a report on the role of B cells and IL-23 in promoting CNS disease in an EAE model. The report demonstrates that IgH[MOG] mice develop exacerbated disease compared to WT after immunization with MOG, with worse clinical disease, demyelination, and accumulation of leukocytes within meningeal immune aggregates. B cells within lesions produce IL-23, and global IL-23 neutralization reduces disease. The authors identify within aggregates the presence of PD-1+ CXCR5- T cells with some features that resemble T peripheral helper cells, a CXCR5- B cell helper T cell population. Few studies in murine models have clearly demonstrated the presence of Tph cells, thus the demonstration of their presence here would be valuable if it can be more convincingly demonstrated.

MAJOR COMMENTS:

1) Tertiary lymphoid structures are often expected to have an organization with distinct T cell and B cell regions, as well as evidence of ongoing B cell activation and somatic hypermutation, as suggested by expression of AID for example. These features distinguish TLS from more loosely organized lymphoid aggregates. Is there evidence of B cell activation or SHM in the meningeal aggregates?

2) Tph cells generally look like Tfh cells except with a different migratory program; the Tfh-like B cell helper program helps to distinguish Tph cells from other, activated PD-1+ T cells. It is currently not convincing that the PD1+ CXCR5- population represents Tph cells rather than activated Th17 cells. Since the authors have transcriptomic data, do the PD1+ CXCR5- cells express a global transcriptomic signature that resembles Tfh cells, and is this different in IgH[MOG] vs WT? How does the transcriptome of PD1+ CXCR5- cells from these mice compare to transcriptomes of bone fide Tfh cells? The selected markers shown (IL21, CX3CR1, CXCL13[discussed below]) are helpful but do not establish the phenotype, especially since ICOS follows the opposite pattern, in contrast to the protein data in Fig5C. It would further support the contention that these are Tph cells if the authors can show that they can provide help to B cells in vitro.

3) Quantification of IL-21 protein by ICS in SuppFig5C is not adequately demonstrated. The authors should show controls of unstimulated cells or stimulated spleen/naïve cells to demonstrate that the shoulder of the contour plot as gated is accurately capturing IL-21 expression. My impression from the plots shown is that IL-21 production is substantially overestimated in this gating/detection. This is an important point because IL-21 is a key factor produced by Tfh and Tph

cells.

4) Expression of CXCL13 is a characteristic feature of human Tfh and Tph cells; yet generally CXCL13 has not been seen as produced by murine T cells. Can the authors show convincingly that T cells (a very large portion of them) produce CXCL13 protein indicated in Fig5C? This would be very surprising but important if correct. Is CXCL13 detectable in supernatants? The RNAseq also suggests CXCL13 expression, but only in Tph from IgH[MOG] and not WT – shouldn't Tph from WT also have some CXCL13 based on Fig5C? Given the value of CXCL13 as a marker of human Tph and Tfh cells, this is an important point to clarify.

5) Detection of p19, p35, p40 in based on example plots in SuppFig4 is not convincing; it is not clear that the authors can really quantify expression of these proteins. The detection of IL-23 heterodimer in Figure 4 seems more robust from the example provided.

MINOR:

1) Figures such as SuppFig5E should show individual datapoints.

2) Quantification of PD1+ CXCR5- cells in spleen in SuppFig5 (showing range of 5-13%) does not match the example flow plots provided, which show far smaller numbers.

Reviewer #3 (Remarks to the Author):

NCOMMS-23-45911

In this manuscript by Fazazi et al. the authors study the IgH[MOG] mouse model, which has ~30% MOG-reactive B cells, particularly assessing the impact of autoreactive B cells on the T cell response. This work touches on an important open question in the field—the mechanism of action of B cell depletion therapies in MS—and the authors present a logical and convincing series of experiments. They find worse EAE in IgH[MOG] mice after MOG35-55 immunization, and show some evidence that this is not due to antibody production. They find increased leukocyte infiltration into the CNS and increased accumulation of leukocytes in the cerebellar leptomeninges. They link the worse disease outcome to production of IL-23 by B cells in the IgH[MOG] mice, leading to a heightened Th17 response. Overall, the main claims of this paper are supported by the presented data. There are a few areas requiring clarification in the text and additional experiments.

- In numerous experiments the authors perform analyses on immunized WT and IgH[MOG] animals, but the time point of these analyses is not clear from the text/legends/methods. For the purposes of this review I am assuming all the experiments, including IHC, flow, RNAseq, MOG IgG serum quantification, etc., were done at the same day post immunization on comparable cohorts. What age/sex were the mice used in the experiments in this study? Were experiments repeated? Please

clarify these details in the manuscript.

- Will RNAseq data be uploaded to a publicly available repository at the time of publication?
- Serum anti-MOG IgG levels are not statistically significantly different (though there appear to be two extreme outliers in the WT group), supporting a non-antibody related mechanism for worse EAE in IgH[MOG] mice. However, the authors also show increased class switching of B cells, as well as increased B cell meningeal infiltrates. To strengthen their conclusion, I suggest assess for Ig deposition in the CNS parenchyma of immunized WT and IgH[MOG] mice. I would also be useful to measure plasma cells in these groups. Finally, the addition of B cell phenotypic (IgD/IgM, markers of maturity) analysis in non-immunized B cell groups would be helpful to assess for differential B cell responses since the baseline characteristics of WT and IgH[MOG] B cells could be different.
- The authors show increased IL-23 production by B cells in IgH[MOG] mice, is this being produced by all B cells or specifically the MOG-reactive cells?
- The identification of meningeal leukocyte accumulations could be of particular interest to the field given the need for models of autoimmune leptomeningeal inflammation. Did the authors see meningeal inflammation in areas besides the cerebellar meninges, such as cortical, spinal cord, or the base of the brain?
- The author looked for Tfh cells in the dura mater by flow, finding no increase in them despite the increased meningeal leukocyte accumulations there. Since the authors show meningeal inflammation in the cerebellar meninges, flow should be done on the leptomeninges there to assess for Tfh, and to further characterize the cellular makeup of leptomeningeal leukocyte accumulations.

Minor issues:

- some instances of “(ref)” in the text — was this intentional?
- line 117 typo
- ‘In vivo’ / ‘In vitro’ should not be italicized
- TNF alpha has been renamed TNF

Peter Calabresi

We sincerely thank the 3 reviewers for their critiques that have resulted in a much stronger manuscript. While we were greatly encouraged by their positive comments, all reviewers indicated that important revisions were necessary: of the 34 requested revisions, 15 required new experiments. We have addressed the vast majority of these latter critiques with new data as outline in our point by point response below.

Reviewer #1:

“In the last several years, B cells have been found to play important roles in MS pathogenesis in addition to T cells, which are largely considered to be the drivers of MS pathophysiology. The success of B cell-depleting therapies has demonstrated their clinical relevance in MS but the biological mechanisms are only partially understood, while many ‘T cell’ therapies also show an impact on B cells. In this paper, Fazazi et al. investigate the mechanisms behind how MOG reactive B cells mediate T cell driven pathogenicity, which has not been fully elucidated before. To do this, the authors induce MOG35-55 active EAE in IgH[MOG] mice, in which the B cells are skewed to recognizing the MOG protein. Using this model, they show that the presence of MOG reactive B cells further exacerbates EAE clinical scores during active EAE without altering peripheral T or B cell activation or MOG specific IgG production. In contrast, in the CNS, they show increased accumulation of tertiary lymphoid organs in the meninges and increased cytokine production by the B cells and Th17 cells of the IgH[MOG] mice. They also show that adoptive transfer of IgH[MOG] B cells in addition to Th17 cells results in exacerbation of EAE clinical scores in the passive EAE model. IL23, a Th17 stabilization factor, has been previously shown to be secreted by B cells, and in their model only B cells (not DCs, astros or microglia) from IgH-MOG secrete higher levels of IL-23 compared to WT. The authors further show that blockade of IL23p19 using a blocking antibody resulted in amelioration of EAE and reduced the presence of TLOs and frequency of T peripheral helper cells in the CNS of the treated mice. They also show a correlation between the proportion of B cells in the CSF of pwMS and levels of IL23 mRNA, suggesting that this mechanism could be at play in MS as well. They finally determined that pathways associated with neurodegeneration and oxidative stress are overrepresented in the IgH[MOG] B cells and T peripheral helper cells compared to wild type, before showing that blockade of IL23 reduces reactive oxygen species production by B cells and Tph cells.

Overall, using multiple EAE models and approaches, the authors convincingly show that B cells facilitate pathogenic Th17 function, suggesting that B cell-Th17 cell interactions are essential in CNS autoimmunity and at least partially dependent on IL-23. The methodology is sound and the methods section is sufficiently detailed. The conclusions made are supported by the evidence shown and the mechanism identified is both relevant (scientifically and clinically) and significant to the field.” We are gratified by the reviewer’s appreciation of the robustness of our findings and their potential impact. Below, we outline how we have addressed their specific concerns.

“Minor concerns to address: Demonstrating that B cells from MS patients in either peripheral blood or CSF produce higher levels of IL-23 by ELISA or express IL-23 more frequently by FACS or mRNA for il23a, generating new data or mining published scRNAseq or bulk RNAseq from B cells of MS vs. healthy controls, or showing B cells expressing IL-23 in the CNS of MS patients, or showing that B cells from MS express lower levels of IL-23 after

induction therapy or B cell-depletion would be an interesting addition to emphasize the importance of the mechanism identified in human MS.” We analyzed publicly available bulk RNA-seq data of B cells from pwMS vs HC¹ (GSE173789) and found that *Il23a* transcript expression trended towards being higher in pwMS ($p_{\text{adj}} < 0.078$). These data are now presented in **Supplementary Fig 4g** and the results are presented on lines **308-311**.

“In Figure 6, it would be more informative to see some of the specific genes associated with oxidative stress and neurodegeneration that are differentially expressed in the IgH[MOG] mice, either as a table of the fold changes and p-values or graphs of the normalized counts.” We agree with the reviewer that these data could have been presented more informatively. We chose to supplement with cnetplot analyses (**Supplementary Fig 7**, described in lines **408-416**) that highlight both fold change as well as genes in common between the GO terms “neurodegeneration” and “oxidative phosphorylation”. All enriched genes had a $p_{\text{adj}} < 0.05$.

“Disease is dependent on IL-23, which is produced by B cells and is required for the augmented Th17 responses seen in these mice.’ As the authors have shown that other immune cells (DCs, monos, macros) also produce IL-23, although there are no differences between WT and IgH-MOG in terms of IL-23 production by these subsets contrarily to what they found in B cells, this statement should be reformulated.” We have reformulated this text (lines **535-537**).

“In general the number of animals, sections, replicates could be more clearly indicated (indicate if n refers to the number of animals, of sections, etc.). Show all points clearly (not overlapped) and describe what the lines are indicating.” n refers to biological replicates. In violin plots/bar graphs, each symbol represents an individual mouse. For histopathological analyses, 3 sections were assessed per mouse (one cervical, one thoracic, one lumbar for spinal cord or at least 3 regions of interest for brain sections). We are now explicit about the number of experimental replicates and apologize that this was not clear before. We have formatted the plots to show points in a “scattered” manner which we hope will make the number of samples per group more evident. Horizontal lines in violin plots represent medians; this is now indicated in each Figure legend.

“For example, Figure 1b: figure legend describes n of 4 but there are only two points for some? please describe the number of animals and the number of sections per animal studied for each region, and ensure that $n \geq 4$.” The number of animals analyzed was $n=4$. This was previously less clear due to the alignment of symbols but we hope it is now evident (**Supplementary Figure 1b** of current manuscript). For histopathological analysis, three sections (one cervical, one thoracic, one lumbar) were analyzed per sample. This is now indicated in the Methods (lines **606-608**).

“Figure 1d: representative of triplicates from how many animals?” Cultures were derived from individual mice; $n=3$ WT and $n=3$ IgH^[MOG] (lines **908-909**).

“Figure 2b: why are there two points next to each other X 3 for IgH-MOG mice (quantification)?” As above, we now present the data (**Figure 2a** of present submission) in scattered format. Further, we more thoroughly assessed lymphocytic clusters throughout the brain in response to a critique from reviewer #3. A heatmap indicating the distribution of lesions in the brain meninges, brain sulci and cerebellar meninges is present in **Supplementary Figure 2b**.

“Figure 4b: please do not use two different shades of blue for two different markers, CD4 expression is difficult to discern. In higher magnification image the cells expressing B220 seem to lack nuclei (DAPI). Please provide new images.” We now provided images that are differently colored in **Figure 4b**: B220 (red), CD4 (aqua), IL-23 (green), DAPI (blue).

“There are multiple places in the manuscript where the reference superscript is preceded by “(ref)”, please fix this typo”. We used this formatting in cases in which the superscript follows a number (e.g., CD20) and might therefore be confused with being an exponent, or in cases in which it follows text that is itself superscripted (e.g., B220^{lo}). We have re-worded to avoid this problem.

“Line 178 a/b typo following PDGFR” This has been corrected.

“Fig. S4C mislabeling?” This has been corrected.

“The font sizes of the axes for many of the graphs are very small and should be increased to help with visibility.” We have endeavored to increase font size in graphs as well as gate enumeration in flow cytometry plots.

“Is there a reason why the first graphs in figure 1 with the EAE curves are split by sex and then this is not done for later analyses? Can you specify this reason in the results and/or discussion and briefly discuss how sex of the mice could or could not play a role in the mechanisms you identified? And what this would imply for MS pathogenesis, where sex is a risk factor depending on the type?” As the reviewer notes, sex is a critical variable in the pathogenesis of MS as well as certain EAE models. Indeed, we recently showed that outcomes in a 1C6 Th17 adoptive transfer model are sex-dependent ². Thus, in **Figure 1**, we compared disease severity between WT and IgH^[MOG] mice of both sexes, to assess whether any differences between the strains might be sex-regulated. However, we found that IgH^[MOG] mice of both sexes developed severe disease. Thus, throughout the paper, we used both male and female mice, as we now clarify in lines **135-136** of the Results. It should be noted that actively immunized NOD mice show no sex difference in EAE ³. We discuss the role of sex in this model briefly in the Discussion (lines **462-468**). In each Figure legend, we are now explicit as to the sex used for each experiment.

“In Figure 6 c-f, combining the graphs of the ROS production in WT & IgH[MOG] and the Isotype and a-IL23 treated mice for each cell type would be more visually striking. If there is a reason to keep them separate, please explain.” In **c** and **d**, we assessed ROS from WT vs IgH^[MOG] mice (B cells, **c**; Tph cells, **d**), while in **e** and **f**, we assessed ROS in B and Tph from IgH^[MOG] treated either with isotype or anti-p19. As the experimental setups were quite different, we feel it is more appropriate to keep **c** distinct from **e**, and **d** distinct from **f**.

“The oxidative stress paragraph in the discussion is minimal, please expand on what is known about oxidative stress in B cells and Tph cells and how this ties into your data and MS pathology.” We have expanded on the discussion of oxidative stress in lines **520-532**.

Reviewer #2:

“Fazazi and colleagues provide a report on the role of B cells and IL-23 in promoting CNS disease in an EAE model. The report demonstrates that IgH[MOG] mice develop exacerbated disease compared to WT after immunization with MOG, with worse clinical disease, demyelination, and accumulation of leukocytes within meningeal immune aggregates. B cells within lesions produce IL-23, and global IL-23 neutralization reduces in disease. The authors identify within aggregates the presence of PD-1+ CXCR5- T cells with some features that resemble T peripheral helper cells, a CXCR5- B cell helper T cell population. Few studies in murine models have clearly demonstrated the presence of Tph cells, thus the demonstration of their presence here would be valuable if it can be more convincingly demonstrated.” We appreciate the reviewer’s assessment of the value of our findings regarding murine Tph cells; nevertheless, they raised important concerns regarding these and other findings that we have addressed as outlined below.

MAJOR COMMENTS

“1) Tertiary lymphoid structures are often expected to have an organization with distinct T cell and B cell regions, as well as evidence of ongoing B cell activation and somatic hypermutation, as suggested by expression of AID for example. These features distinguish TLS from more loosely organized lymphoid aggregates. Is there evidence of B cell activation or SHM in the meningeal aggregates?” We now present immunofluorescence images displaying CD138 (terminally differentiated plasma cells) and AID positivity (class switching/somatic hypermutation) mainly located in meningeal sections (**Figure 2c**, lines 193-198).

Multiple studies have shown the presence of tertiary lymphoid structures in EAE ⁴⁻⁷. In a recent study, Bhargava and colleagues ⁷ investigated the presence of meningeal aggregates in an EAE model in detail. The spatial distribution of B cells and T cells in our samples is reminiscent of what was observed in this study; further, they, like us, saw evidence of fibrinogen deposition under the lymphocytic aggregates, as evidence of more organized lymphoid aggregates. Barghava et al. refer to these structures as being lymphocytic aggregates with “features of tertiary lymphoid tissue”; following this lead, throughout our manuscript, we now use similar wording to refer to the structures that we observe. It should also be noted that a recent paper ⁸ confirmed the presence of organized lymphoid structures in the meninges that, like ours, showed clusters of B cells (including CD138⁺) and T cells with a small presence of CD11c⁺ DC.

“2) Tph cells generally look like Tfh cells except with a different migratory program; the Tfh-like B cell helper program helps to distinguish Tph cells from other, activated PD-1+ T cells. It is currently not convincing that the PD1+ CXCR5- population represents Tph cells rather than activated Th17 cells. Since the authors have transcriptomic data, do the PD1+ CXCR5- cells express a global transcriptomic signature that resembles Tfh cells, and is this different in IgH[MOG] vs WT? How does the transcriptome of PD1+ CXCR5- cells from these mice compare to transcriptomes of bone fide Tfh cells? The selected markers shown (IL21, CX3CR1, CXCL13[discussed below]) are helpful but do not establish the phenotype, especially since ICOS follows the opposite pattern, in contrast to the protein data in Fig5C. It would

further support the contention that these are Tph cells if the authors can show that they can provide help to B cells in vitro.”

We agree with the reviewer that our conclusions regarding Tph cells could be more firmly grounded and appreciate these suggestions, which we have endeavored to incorporate in full:

Bioinformatics: With our bioinformatics collaborators Charles Joly-Beauparlant and Arnaud Droit, we exploited publicly available -seq data derived from murine Tfh and non-Tfh cells⁹ and calculated the Euclidean distance between the centroids of these two populations and the transcriptomes of our Tph-like cells. In doing so, we found that the transcriptomes of both WT and IgH^[MOG] Tph-like cells are closer to those of Tfh cells than to non-Tfh (**Supplementary Figure 6a**, lines 364-369).

Flow cytometry: We have enhanced our flow data by *i*) including an additional marker, MHC class II, that was identified by Rao and Brenner¹⁰ as marking human Tph cells (**Figure 5c**, **Supplementary Figures 5c, e**); *ii*) assessing the expression of these markers on PD-1⁺ vs PD-1^{neg} cells in the spleen (ie. immune periphery) in addition to the CNS (**Supplementary Figure 5e**); *iii*) presenting representative plots (**Supplementary Figures 5c, e**) and fluorescence minus one (FMO) controls (**Supplementary Figure 5e**) that were lacking in the first submission (discussed further below).

B cell help: Adapting a previously described in vitro T:B cell help assay that has been used to validate Tfh differentiation¹¹, we find that PD1⁺CXCR5⁻ T cells from both immunized WT and IgH^[MOG] mice are better able to elicit an IgM^{lo}IgD^{lo} phenotype from responder B cells relative to PD1^{neg} counterparts from the same mice (**Supplementary Figure 6d**, lines 376-381).

Interpretation: Notwithstanding the above, throughout the text we have additionally softened our description of CD4⁺PD1⁺CXCR5⁻ cells to be “Tph-like” or “resembling Tph”. We feel that this may be more prudent, given that evidence of Tph cells in murine cells in the literature is, thus far, quite limited.

“3) Quantification of IL-21 protein by ICS in SuppFig5C is not adequately demonstrated. The authors should show controls of unstimulated cells or stimulated spleen/naïve cells to demonstrate that the shoulder of the contour plot as gated is accurately capturing IL-21 expression. My impression from the plots shown is that IL-21 production is substantially overestimated in this gating/detection. This is an important point because IL-21 is a key factor produced by Tfh and Tph cells.” We now present FMO controls for IL-21 from CNS T cells (**Supplementary Figure 5c**), as well as staining data from splenic T cells (**Supplementary Figure 5e**) that are essentially negative for IL-21. These demonstrate that our gating is indeed capturing IL-21⁺ cells in the CNS compartment.

“4) Expression of CXCL13 is a characteristic feature of human Tfh and Tph cells; yet generally CXCL13 has not been seen as produced by murine T cells. Can the authors show convincingly that T cells (a very large portion of them) produce CXCL13 protein indicated in Fig5C? This would be very surprising but important if correct. Is CXCL13 detectable in supernatants? The RNAseq also suggests CXCL13 expression, but only in Tph from IgH[MOG] and not WT – shouldn't Tph from WT also have some CXCL13 based on Fig5C?”

Given the value of CXCL13 as a marker of human Tph and Tfh cells, this is an important point to clarify.” It was indeed proposed by Crotty¹² that murine Tfh cells do not produce CXCL13. However, two recent papers^{13,14} showed that CXCL13 can indeed be produced by murine T cells in vivo. The former¹³, in *Immunity*, demonstrated this fact by both flow cytometry and immunofluorescence. We discuss these studies on lines **513-519**.

Nonetheless, we agree with the reviewer that it is important to demonstrate this point more conclusively in our own study and have done so via the following: *i*) we now show representative plots and FMO controls (**Supplementary Figure 5c**) – we acknowledge that these would have been helpful in the first submission and apologize for not including them; *ii*) direct comparison of expression of CXCL13 between WT and IgH^[MOG] PD1⁺ cells, showing that it is most highly expressed in IgH^[MOG] PD1⁺ cells as compared to either WT PD1⁺ and IgH^[MOG] PD1^{neg} cells (**Fig 5c**); *iii*) following the example of Chaurio et al¹³, we found evidence of CXCL13 and CD4 colocalization in the meninges by immunofluorescence (**Supplementary Figure 5d**). These data, in addition to our finding that CXCL13 is upregulated by IgH^[MOG] Tph-like cells at the transcriptional level (**Supplementary Figure 6c**), collectively support the observation that IgH^[MOG] Tph-like cells produce CXCL13.

“5) Detection of p19, p35, p40 in based on example plots in SuppFig4 is not convincing; it is not clear that the authors can really quantify expression of these proteins. The detection of IL-23 heterodimer in Figure 4 seems more robust from the example provided.” In the first submission, we presented evidence that IgH^[MOG] B cells can express IL-23 using a heterodimer-specific antibody, but also data of co-expression of p19 and p40 (IL-23 chains) as well as p35 and p40 (IL-12 chains). In reflecting on the reviewer’s critique, we determined that (outside of any issues with the staining), the p19/p35/p40 monomeric co-expression data did not conclusively demonstrate the stated conclusions, as co-expression of p19 and p40 does not necessarily prove that these cells express IL-23 as a heterodimer; similarly, co-expression of p35 and p40 does not permit us to conclude that IL-12 is present as a heterodimer. We therefore removed these data.

Instead, we show, in **Supplementary Figure 4c**, that p35 monomer is not expressed in B cells of either genotype; we can therefore infer that they do not express IL-12 heterodimer. Further, in **Supplementary Figure 4d**, we have replaced the p19/p40 co-expression data from other cell types with expression of the IL-23 heterodimer. We thank the reviewer for this critique that led to our reconsideration of how we present this data.

MINOR COMMENTS:

“1) Figures such as SuppFig5E should show individual datapoints.” We have reformatted these plots to show individual data points (now **Supplementary Figure 6c**) for *Cxcl13*, *Il17a* and *Il17f*. Of note, the *Cxcl13* data further support our finding that IgH^[MOG] PD1⁺CXCR5⁻ cells show increased expression of this marker.

“2) Quantification of PD1+ CXCR5- cells in spleen in SuppFig5 (showing range of 5-13%) does not match the example flow plots provided, which show far smaller numbers.” In the first submission, we had mislabeled the axes, switching PD-1 (should be X) and CXCR5 (should be Y). This has now been fixed. We thank the reviewer for catching this error.

Reviewer #3:

“In this manuscript by Fazazi et al. the authors study the IgH[MOG] mouse model, which has ~30% MOG-reactive B cells, particularly assessing the impact of autoreactive B cells on the T cell response. This work touches on an important open question in the field—the mechanism of action of B cell depletion therapies in MS—and the authors present a logical and convincing series of experiments. They find worse EAE in IgH[MOG] mice after MOG35-55 immunization, and show some evidence that this is not due to antibody production. They find increased leukocyte infiltration into the CNS and increased accumulation of leukocytes in the cerebellar leptomeninges. They link the worse disease outcome to production of IL-23 by B cells in the IgH[MOG] mice, leading to a heightened Th17 response. Overall, the main claims of this paper are supported by the presented data. There are a few areas requiring clarification in the text and additional experiments.” We thank the reviewer for their comments regarding the robustness of our findings and below, we outline how we have addressed their concerns regarding the manuscript.

“In numerous experiments the authors perform analyses on immunized WT and IgH[MOG] animals, but the time point of these analyses is not clear from the text/legends/methods. For the purposes of this review I am assuming all the experiments, including IHC, flow, RNAseq, MOG IgG serum quantification, etc., were done at the same day post immunization on comparable cohorts. What age/sex were the mice used in the experiments in this study? Were experiments repeated? Please clarify these details in the manuscript.” We apologize for the lack of clarity. After the initial analyses of disease evolution (**Figure 1a**), we defined endpoints (for both WT and IgH^[MOG]) as being when IgH^[MOG] mice attained a disease score of 3 or greater for 3 consecutive days. This was done in part to assuage the concerns of our veterinary service regarding the monitoring of very sick mice for lengthy periods of time. WT mice never attained disease of this severity in the same time scale as IgH^[MOG] mice. Mice were used between 9-12 weeks of age. We now elaborate on these details in the Methods (**lines 576-584**). We now indicate more clearly the number of experimental repeats in the figure legends.

“Will RNAseq data be uploaded to a publicly available repository at the time of publication?” The data have been uploaded to GEO (#PRJNA1021552) and will be released upon manuscript acceptance.

“Serum anti-MOG IgG levels are not statistically significantly different (though there appear to be two extreme outliers in the WT group), supporting a non-antibody related mechanism for worse EAE in IgH[MOG] mice. However, the authors also show increased class switching of B cells, as well as increased B cell meningeal infiltrates. To strengthen their conclusion, I suggest assess for Ig deposition in the CNS parenchyma of immunized WT and IgH[MOG] mice.” While we agree that the two WT samples appear by eye to be outliers, upon running Grubb’s test on the data, we found that they do not meet the statistical definition. We assessed three regions of the brain parenchyma in WT (n=5) and IgH^[MOG] (n=6) mice for IgG deposition via immunofluorescence. We found higher IgG deposition in IgH^[MOG] samples (**Supplementary Figure 2d**, lines 198-200), suggesting a pathological role of IgG in our model.

“It would also be useful to measure plasma cells in these groups.” We now present immunofluorescence images displaying CD138 (plasma cell marker) in meningeal sections (**Figure 2c**, lines 193-196).

“Finally, the addition of B cell phenotypic (IgD/IgM, markers of maturity) analysis in non-immunized B cell groups would be helpful to assess for differential B cell responses since the baseline characteristics of WT and IgH[MOG] B cells could be different.” Data for splenic mature, immature and CS B cells from unimmunized mice are now presented in **Supplementary Figure 3b** with results presented on lines 216-220.

“The authors show increased IL-23 production by B cells in IgH[MOG] mice, is this being produced by all B cells or specifically the MOG-reactive cells?” To get at this question, we used streptavidin-labeled MOG to identify MOG-specific B cells and first assessed the distribution of MOG-specific vs. non-specific B cells in various tissues. As expected, a strong proportion of B cells in the immune periphery (spleen, cervical LN) of IgH^[MOG] mice were MOG-specific (**Supplementary Figure 3e**). Among splenic IgH^[MOG] B cells, IL-23 expression was significantly higher in the MOG-nonspecific fraction (**Supplementary Figure 4b**), though it should be noted that the expression of IL-23 from B cells in this compartment is modest.

Intriguingly, in the course of these investigations, we found that MOG reactive B cells are excluded from the CNS parenchyma (**Supplementary Figure 3f**). While this may appear surprising on the surface, our findings match the previous observations of Tesfagiorgis¹⁵ and Wang¹⁶. MOG-reactive B cells are also lacking in the subdural meninges (SDM) but, interestingly, do arise in the dura mater of immunized IgH^[MOG] mice (**Supplementary Figure 3f**).

“The identification of meningeal leukocyte accumulations could be of particular interest to the field given the need for models of autoimmune leptomeningeal inflammation. Did the authors see meningeal inflammation in areas besides the cerebellar meninges, such as cortical, spinal cord, or the base of the brain?” Our updated quantification of leukocytic aggregations now takes into account the brain meninges, brain sulci and cerebellar meninges (**Figure 2a**), which are the 3 areas in which we identified these structures. We present a breakdown of these localizations in **Supplementary Figure 2b**. In our histopathological workflow, we used brains to generate frozen sections (for IF) and used cord to generate paraffin blocks. Thus, we could not carry out a comparable assessment of spinal cord sections based on the samples we had in hand. We are now explicit (lines 620-623) that we studied brains alone.

“The author looked for T_{fh} cells in the dura mater by flow, finding no increase in them despite the increased meningeal leukocyte accumulations there. Since the authors show meningeal inflammation in the cerebellar meninges, flow should be done on the leptomeninges there to assess for T_{fh}, and to further characterize the cellular makeup of leptomeningeal leukocyte accumulations.” We now show that T_{fh} cells are increased in the subdural meninges (**Figure 5e**). In addition, we determined that MOG-specific B cells can be found in the dura mater as opposed to SDM (**Supplementary Figure 3f**).

“Minor issues: some instances of “(ref)” in the text — was this intentional?” We used this formatting in cases in which the superscript follows a number (e.g., CD20) and might therefore

be confused with being an exponent, or in which it follows text that is itself superscripted (e.g., B220^{lo}). We have re-worded where needed to avoid this problem.

“line 117 typo” We have fixed this typo.

“‘In vivo’ / ‘In vitro’ should not be italicized”. This has been fixed throughout.

“TNF alpha has been renamed TNF” This has been fixed throughout.

References:

1. Ma, Q. *et al.* Specific hypomethylation programs underpin B cell activation in early multiple sclerosis. *Proc. Natl. Acad. Sci. U.S.A.* **118**, e2111920118 (2021).
2. Doss, P. M. I. A. *et al.* Male sex chromosomal complement exacerbates the pathogenicity of Th17 cells in a chronic model of central nervous system autoimmunity. *Cell Rep* **34**, 108833 (2021).
3. Papenfuss, T. L. *et al.* Sex differences in experimental autoimmune encephalomyelitis in multiple murine strains. *J. Neuroimmunol.* **150**, 59–69 (2004).
4. Peters, A. *et al.* Th17 cells induce ectopic lymphoid follicles in central nervous system tissue inflammation. *Immunity* **35**, 986–996 (2011).
5. Pikor, N. B. *et al.* Integration of Th17- and Lymphotoxin-Derived Signals Initiates Meningeal-Resident Stromal Cell Remodeling to Propagate Neuroinflammation. *Immunity* **43**, 1160–1173 (2015).
6. Ward, L. A. *et al.* Siponimod therapy implicates Th17 cells in a preclinical model of subpial cortical injury. *JCI Insight* **5**, (2020).
7. Bhargava, P. *et al.* Imaging meningeal inflammation in CNS autoimmunity identifies a therapeutic role for BTK inhibition. *Brain* **144**, 1396–1408 (2021).
8. Fitzpatrick, Z. *et al.* Venous-plexus-associated lymphoid hubs support meningeal humoral immunity. *Nature* 1–8 (2024). doi:10.1038/s41586-024-07202-9
9. Kumar, P. *et al.* Restoration of Follicular T Regulatory/Helper Cell Balance by OX40L-JAG1 Cotreatment Suppresses Lupus Nephritis in NZBWF1/j Mice. *J. Immunol.* **208**, 2467–2481 (2022).
10. Rao, D. A. *et al.* Pathologically expanded peripheral T helper cell subset drives B cells in rheumatoid arthritis. *Nature* **542**, 110–114 (2017).
11. Gao, X., Wang, H., Chen, Z., Zhou, P. & Yu, D. An optimized method to differentiate mouse follicular helper T cells in vitro. *Cell. Mol. Immunol.* **17**, 779–781 (2020).
12. Crotty, S. Follicular helper CD4 T cells (TFH). *Annu. Rev. Immunol.* **29**, 621–663 (2011).
13. Chaurio, R. A. *et al.* TGF- β -mediated silencing of genomic organizer SATB1 promotes Tfh cell differentiation and formation of intra-tumoral tertiary lymphoid structures. *Immunity* **55**, 115–128.e9 (2022).
14. Guo, Z. *et al.* IL-10 Promotes CXCL13 Expression in Macrophages Following Foot-and-Mouth Disease Virus Infection. *Int J Mol Sci* **24**, 6322 (2023).
15. Tesfagiorgis, Y., Zhu, S. L., Jain, R. & Kerfoot, S. M. Activated B Cells Participating in the Anti-Myelin Response Are Excluded from the Inflamed Central Nervous System in a Model of Autoimmunity that Allows for B Cell Recognition of Autoantigen. *J. Immunol.* **199**, 449–457 (2017).

16. Wang, Y. *et al.* Early developing B cells undergo negative selection by central nervous system-specific antigens in the meninges. *Immunity* **54**, 2784–2794.e6 (2021).

REVIEWERS' COMMENTS

Reviewer #1 (Remarks to the Author):

All my concerns have been well addressed in the revisions. I recommend to accept the manuscript, which provides novel scientifically and clinically relevant data demonstrating how B cells can contribute to sustained neuroinflammation in MS by recruiting pathogenic Th17 cells and increasing oxidative stress.

Reviewer #2 (Remarks to the Author):

I am satisfied with the authors' revisions and additions. The case has been substantially strengthened that the cells described here should be considered Tph cells. This is quite valuable.

One comment: I still have some skepticism about the accuracy of the quantification of CXCL13 in Fig6 by ICS. Comparison to FMO is helpful but does not account for non-specific staining, which is particularly relevant for intracellular stains. Comparison to spleen is also valuable, yet it seems odd that splenic T cells make CXCL13 but not IL-21 (are there Tfh cells there, or not?). A clear demonstration of production CXCL13 in supernatants would provide a more definitive demonstration that murine T cells can produce this chemokine at reasonable levels. The papers cited that mouse T cells make CXCL13 also largely rely on ICS, carrying the same ambiguity as the current paper; it risks building a case in the literature for a murine T cell function that doesn't actually occur at meaningful levels. Nonetheless, this is not a critical point for this manuscript, and I am swayed by the RNA-seq data showing CXCL13 expression in IgH[MOG] cells; I think this is sufficient for this manuscript.

Reviewer #3 (Remarks to the Author):

The authors have adequately responded to all of my concerns.

We greatly appreciate the comments of the 3 reviewers as pertains to our resubmission of this manuscript. Our point-by-point response is below:

Reviewer #1:

“All my concerns have been well addressed in the revisions. I recommend to accept the manuscript, which provides novel scientifically and clinically relevant data demonstrating how B cells can contribute to sustained neuroinflammation in MS by recruiting pathogenic Th17 cells and increasing oxidative stress.”

We thank the reviewer for their appreciation of our revised manuscript.

Reviewer #2:

“I am satisfied with the authors' revisions and additions. The case has been substantially strengthened that the cells described here should be considered Tph cells. This is quite valuable.”

We thank the reviewer for their appreciation of our efforts to address their concerns. They raised valuable critiques regarding the phenotyping of Tph cells that helped to strengthen the manuscript.

“One comment: I still have some skepticism about the accuracy of the quantification of CXCL13 in Fig6 by ICS. Comparison to FMO is helpful but does not account for non-specific staining, which is particularly relevant for intracellular stains. Comparison to spleen is also valuable, yet it seems odd that splenic T cells make CXCL13 but not IL-21 (are there Tfh cells there, or not?). A clear demonstration of production CXCL13 in supernatants would provide a more definitive demonstration that murine T cells can produce this chemokine at reasonable levels. The papers cited that mouse T cells make CXCL13 also largely rely on ICS, carrying the same ambiguity as the current paper; it risks building a case in the literature for a murine T cell function that doesn't actually occur at meaningful levels. Nonetheless, this is not a critical point for this manuscript, and I am swayed by the RNA-seq data showing CXCL13 expression in IgH[MOG] cells; I think this is sufficient for this manuscript.”

In response to this critique, we have modified our discussion of CXCL13 in the discussion to add the proviso, “further work is needed to define the functional significance of these observations.”

Reviewer #3:

“The authors have adequately responded to all of my concerns.”

We thank the reviewer for their appreciation of our revised manuscript.